# Complementary codes for odor identity and intensity in olfactory cortex

Kevin A Bolding, Kevin M Franks*

Department of Neurobiology, Duke University Medical School, Durham, United States

**Abstract** The ability to represent both stimulus identity and intensity is fundamental for perception. Using large-scale population recordings in awake mice, we find distinct coding strategies facilitate non-interfering representations of odor identity and intensity in piriform cortex. Simply knowing which neurons were activated is sufficient to accurately represent odor identity, with no additional information about identity provided by spike time or spike count. Decoding analyses indicate that cortical odor representations are not sparse. Odorant concentration had no systematic effect on spike counts, indicating that rate cannot encode intensity. Instead, odor intensity can be encoded by temporal features of the population response. We found a subpopulation of rapid, largely concentration-invariant responses was followed by another population of responses whose latencies systematically decreased at higher concentrations. Cortical inhibition transforms olfactory bulb output to sharpen these dynamics. Our data therefore reveal complementary coding strategies that can selectively represent distinct features of a stimulus.

## Introduction

Olfaction plays a central role in finding food, avoiding predators and selecting a mate, and is thus crucial for survival. These behaviors require that an animal can extract and represent different features of the odor stimulus. For example, an animal following a scent trail must be able to reliably recognize the target odor over a large range of concentrations. However, the animal must also be able to discriminate small changes in odorant concentration if it is to track the scent to its source. This presents a challenge for the olfactory system because it must form a representation of odor identity that is robust to changes in concentration while simultaneously retaining the ability to represent odor intensity.

Odors are detected by an array of odorant receptors on olfactory sensory neurons in the nasal epithelium. Each olfactory sensory neuron expresses just one type of odorant receptor, and all sensory neurons expressing a given receptor project to a unique pair of glomeruli in the olfactory bulb, so that odors are first represented by combinations of active glomeruli that each represent elemental molecular features of the odorant (*Wilson and Mainen, 2006*). This information is then projected by olfactory bulb mitral/tufted cells to piriform cortex, where it is integrated and synthesized, and thought to give rise to the perception of 'odor objects' that should represent different features of the odor, including its identity and intensity (*Gottfried, 2010*; *Wilson and Sullivan, 2011*).

Odors activate distributed and overlapping ensembles of piriform neurons, with a small fraction of cells responding strongly to any given stimulus (*Illig and Haberly, 2003*; *Rennaker et al., 2007*; *Poo and Isaacson, 2009*; *Stettler and Axel, 2009*). Studies in awake, behaving rodents have shown that different odorants can be accurately decoded from populations of piriform neurons using spike counts across the population (i.e. a rate code), with little additional information provided by spike time (*Miura et al., 2012*). These decoding analyses only examined piriform responses to different

*For correspondence: franks@ neuro.duke.edu

**Competing interests:** The authors declare that no competing interests exist.

odorants at a single concentration. However, odor identity, as one feature of the odor stimulus, should be largely independent of odorant concentration. Thus, it is unclear which features of the cortical response represent odor identity per se.

While input to olfactory bulb glomeruli is strongly concentration-dependent (*Rubin and Katz, 1999*; *Spors and Grinvald, 2002*), olfactory bulb output may be normalized over a large range of concentrations (*Banerjee et al., 2015*; *Sirotin et al., 2015*; *Economo et al., 2016*; *Roland et al., 2016*). Therefore, it is unclear how, or even if, odor concentration is represented in piriform cortex (*Sugai et al., 2005*; *Stettler and Axel, 2009*; *Xia et al., 2015*). Odors typically retain their perceptual identities across a range of concentrations (*Krone et al., 2001*; *Laing et al., 2003*; *Homma et al., 2009*). It may be the case that piriform odor representations are largely concentration-invariant. Alternatively, identity and intensity could be represented by different and independent features of the neural response. No previous studies have attempted to decode odor concentration from neural activity in piriform cortex. Therefore, three fundamental questions remain unanswered: how is odor identity represented in piriform cortex, how is odor intensity represented in piriform cortex, if at all, and how do piriform representations of odor identity depend on odorant concentration?

A dissociation of temporal and rate codes in piriform cortex could give rise to non-interfering representations of identity and intensity (*Hopfield, 1995*; *Schaefer and Margrie, 2007*). Odors activate different olfactory bulb glomeruli and mitral/tufted cells at specific phases of the respiration cycle (*Spors et al., 2006*; *Bathellier et al., 2008*; *Cury and Uchida, 2010*; *Shusterman et al., 2011*). Increasing odorant concentration typically does not increase spiking systematically in responsive mitral/tufted cells, but rather decreases their onset latencies (*Chalansonnet and Chaput, 1998*; *Cang and Isaacson, 2003*; *Margrie and Schaefer, 2003*; *Fukunaga et al., 2012*; *Sirotin et al., 2015*). Therefore, the same subset of mitral/tufted cells may be activated by a given odorant at different concentrations, but with shorter latencies at higher concentrations. Thus, if different features of the piriform response are sensitive to the spatial and the temporal (*Haddad et al., 2013*) patterns of its input from olfactory bulb, then these features could independently represent odor identity and odor intensity.

To test these predictions we recorded odor-evoked spiking activity in populations of piriform cortex neurons in awake, head-fixed mice. We found that different odorants activated distinct ensembles of piriform neurons but neither ensemble size nor spike counts in responsive neurons were systematically concentration-dependent, indicating that a rate code cannot be used to represent odor intensity. Instead, we find that responses occur in two phases that become more synchronous at higher concentrations. These concentration-dependent dynamics are not simply inherited from olfactory bulb; intracortical inhibition is sharpened at higher concentrations, suggesting its role in actively transforming input from bulb. Using a linear decoder to classify single-trial responses, we show that odor identity is accurately encoded by the ensemble of activated neurons. By contrast, intensity is represented by the latency of neurons that respond more slowly. We therefore propose that odor identity and odor intensity are represented using distinct and non-interfering coding strategies in piriform cortex.

## Results

We recorded extracellular spiking activity in large populations of anterior piriform cortex neurons in awake, head-fixed mice (*Figure 1A*). Probes were positioned to span layer II of anterior piriform cortex, with the most ventral electrodes in layer I and the most dorsal electrodes in layer III (*Figure 1B*). Our dataset contains 459 layer II neurons from nine separate recordings in five mice (*Figure 1C*; 33–73 well-isolated units per recording, see Methods and *Figure 1—figure supplement 1A–E*). In a subset of these experiments (6/9) we obtained simultaneous recordings from olfactory bulb cells (126 neurons from ventrolateral mitral cell layer, see Materials and methods). Piriform cortex cells typically had low spontaneous firing rates with a log-normal distribution (mean: 3.09 Hz; median: 1.68 Hz; st. dev: 3.93 Hz; range: 0.0028–40.8 Hz; *Figure 1—figure supplement 1F*). Individual cells exhibited weak phase preferences that were uniformly distributed across the sniff so that, unlike in anesthetized conditions (*Rennaker et al., 2007*; *Poo and Isaacson, 2009*), population spiking was decoupled from respiration (*Figure 1—figure supplement 1G*). We estimate that ~93% of these units were principal neurons (see Figure 9), which is consistent with histological measures

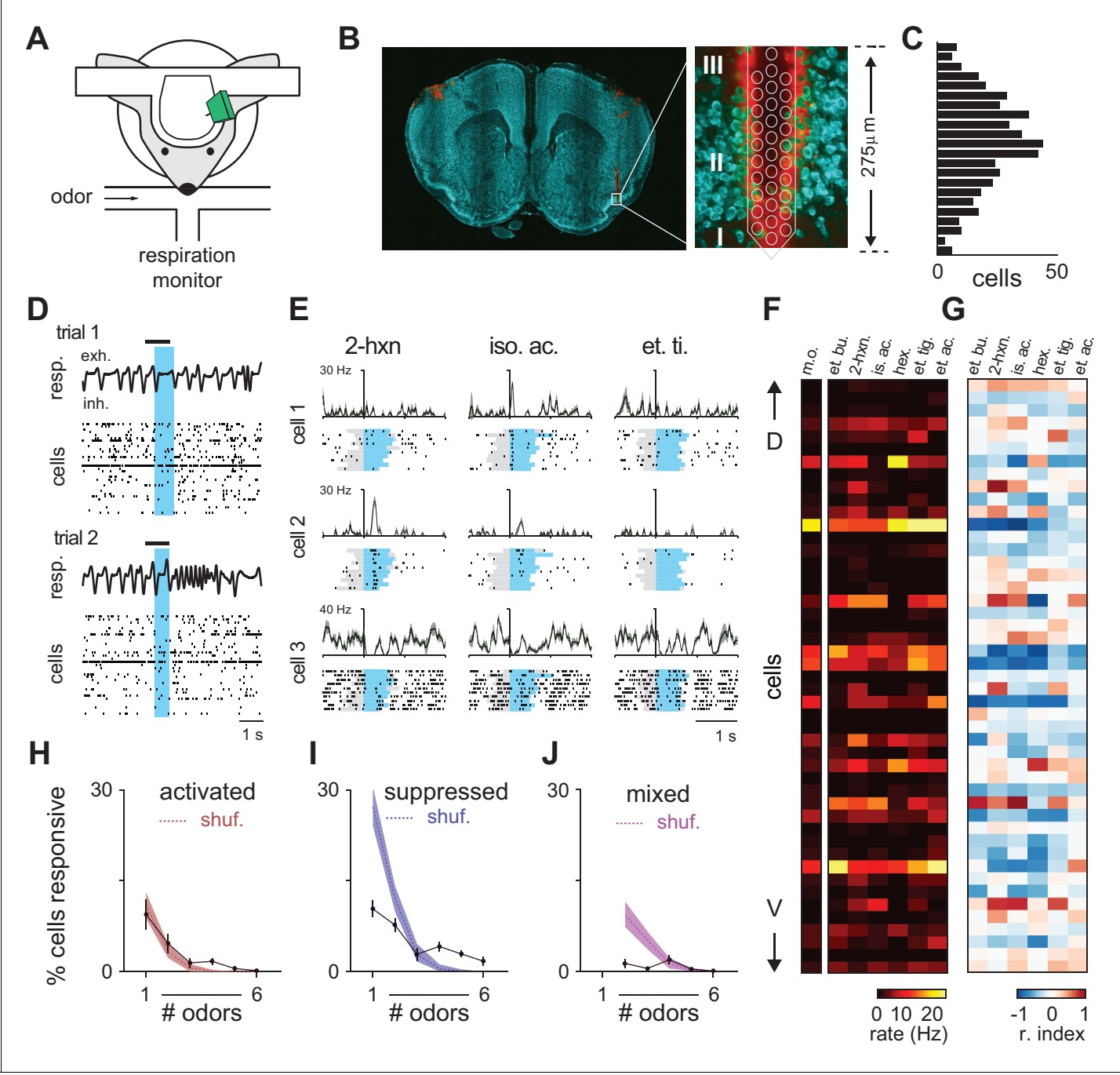

**Figure 1.** Odor responses in piriform cortex. (A) Schematic of experimental setup. (B) Left, Coronal brain section showing region of anterior piriform cortex where recordings were obtained. Multielectrode probes were painted with DiI to mark recording locations (red, DiI; green, NeuroTrace). DiI in the left hemisphere is from a separate recording. Right, higher magnification image of boxed area on left, superimposed with a schematic at scale of a 32-channel probe. Numerals indicate piriform layers. Note the electrode tip spans layer II. (C) Dorso-ventral (DV) distribution of 459 isolated cells (n = 9 experiments) determined by measuring the center of mass of each unit's waveform amplitudes recorded on different channels across the probe. (D) Population activity and timing of odor delivery. One-second odor pulses (black bar, isoamyl acetate) were triggered on exhalation. Blue shaded region indicates first full respiration cycle after odor onset. Population raster plots below display spiking activity of 33 simultaneously recorded cells sorted by estimated DV position within layer II. (E) Inhalation-aligned raster plots showing responses of three simultaneously recorded cells to three different odor stimuli (0.3% v./v.). Gray shading indicates odor delivery. Blue shading indicates respiration cycle. Cells displayed varying degrees of odor specificity, activation or suppression, and temporal precision. (F–G) Population responses in a representative experiment with 48 simultaneously recorded neurons sorted by relative DV location responding to six different monomolecular odorants (0.3%). (F). Firing rates during the first respiration cycle after odor onset. A mineral oil control is shown on left. (G). Same data expressed using a response index where cells with response index values of −1 (blue) and 1

*Figure 1 continued on next page*

Figure 1 continued

(red) are unambiguously suppressed or activated, respectively. Odors: et. bu., ethyl butyrate; 2-hxn., 2-hexanone; is. ac., isoamyl acetate; hex., hexanal; et. ti., ethyl tiglate; et. ac., ethyl acetate. (H–J) Percent cells responding to different numbers of odorants with significantly increased spiking (H), decreased spiking (I), or mixed polarity responses (J). Filled circles are mean ± s.e.m. (n = 9). The dashed line denotes the expected distribution when cell identities are shuffled; shaded area, 5th–95th percentiles.

The following figure supplements are available for figure 1:

**Figure supplement 1.** Spike sorting and spontaneous activity.

**Figure supplement 2.** Characterization of piriform odor responses.

**Figure supplement 3.** Properties of Activated and Suppressed cells.

(*Suzuki and Bekkers, 2010*). These results indicate that our recordings sampled large and relatively unbiased populations of neurons. We first recorded cortical responses evoked by different monomolecular odorants at a nominal concentration (0.3% v./v.) and examined spiking activity across the population for the first sniff after odor onset (*Figure 1D*). While animals typically sniffed actively after odor offset (*Figure 1D*), respiration barely changed during the odor presentation (see *Figure 2—figure supplement 1C*), indicating that odor-evoked changes we observed in neural activity were not complicated by changes in sniffing. We aligned trials to the onset of inhalation. Some cells had very selective, reliable and precise responses (*Figure 1E*, cell 1) while others were somewhat less selective or precise (*Figure 1E*, cell 2), and many cells were suppressed by odor (*Figure 1E*, cell 3). To examine population activity, we constructed response vectors by counting the number of spikes each cell fired in a single sniff. Averaged responses to different odorants for one simultaneously recorded population of neurons are shown as firing rates (*Figure 1F*) and using a response index that quantifies discriminability of odor responses versus blank (*Figure 1G*). A response (i.e. a cell-odor pair response) was defined as 'activated' or 'suppressed' if spiking was statistically significantly higher or lower over the first full respiration cycle compared to a mineral oil 'blank' (p<0.05, rank-sum test). Using this strict criterion, 23% of cells (105/459 cells) were activated by at least one of the six odors, while 33% of cells (150/459 cells) were suppressed, and on average, each odor activated and suppressed 6.7% and 13.3% of cells, respectively (*Figure 1—figure supplement 2*). (These data are largely consistent with previous studies (*Rennaker et al., 2007*; *Poo and Isaacson, 2009*; *Stettler and Axel, 2009*; *Zhan and Luo, 2010*; *Miura et al., 2012*).

## Functionally distinct subpopulations of piriform neurons

However, we observed considerable trial-to-trial variability across the population following repeated presentations of the same odorant. The ability to resolve some cells with exquisitely reliable and precise responses (e.g. cell 1 in *Figure 1*) indicates that this variability cannot be explained by poor stimulus control or misaligned responses, but rather is a true reflection of the population response which could impact odor recognition and discrimination. To quantify that we computed correlations between single-trial response vectors for repeated presentations of the same odorant (0.48 ± 0.12; n = 6 odors, 9 recordings). We then compared these to responses to different odors. Inter-odorant responses were significantly lower, but nevertheless much higher than expected by chance (0.38 ± 0.09, p<0.001, paired t-test).

Olfactory bulb projections to piriform cortex show no discernible topography. If each piriform cell was driven by solely by inputs from a random subset of glomeruli, their responses to different odors would be independent. In this case, we would expect zero correlations between responses to different odors. Why then are responses to different odors so highly correlated? If ensembles were independent we would expect many cells to be activated by some odors and suppressed by others. Instead, we found overrepresented populations of cells that were either broadly activated or broadly suppressed. (*Figure 1H,I*, *Figure 1—figure supplement 3A*). Moreover, only 4% of cells (18/459 cells) showed mixed responses, whereas the distribution of mixed responses predicted from independent ensembles would be 14% (66/459; *Figure 1J*, *Figure 1—figure supplement 3B*). Our data therefore suggest that there are functionally distinct subclasses of piriform neurons that encode

odor stimuli exclusively through either activation or suppression, consistent with a recent imaging study (*Otazu et al., 2015*). We looked for other properties that could distinguish activated and suppressed cells but found no differences in their waveforms or sublaminar positions (*Figure 1—figure supplement 3C–E*). Future experiments will therefore be required to distinguish and further characterize these two cell types.

## Decorrelation and normalization across concentration in bulb and cortex

Odors could retain their perceptual identities over a range of concentrations, if cortical odor responses were concentration-invariant. In fact, it remains unclear whether or how output from olfactory bulb depends on concentration. To address this question we compared responses across odorants or across concentrations of a given odorant (0.03–1%, *Figure 2—figure supplement 1*) in simultaneously recorded populations of olfactory bulb mitral cells and piriform cortex neurons (*Figure 2A*). Trial-to-trial correlations within and across stimuli were lower in piriform cortex than in olfactory bulb (*Figure 2B*). Notably, correlations in both olfactory bulb and piriform cortex decreased systematically with differences in concentration, becoming as dissimilar over a 30-fold concentration range as responses to two different odorants. This analysis reveals that odor representations in both olfactory bulb and piriform cortex, defined by spike counts, are not concentration invariant.

We next asked how odor intensity is encoded in piriform cortex. Intensity may be encoded by a simple, systematic concentration-dependent increase in spiking. This type of cortical rate code would predict that spiking output from olfactory bulb also increased. However total bulb spiking remained unchanged across concentrations (*Figure 2C*), indicating that substantial normalization occurs in olfactory bulb, consistent with recent imaging (*Banerjee et al., 2015*; *Economo et al., 2016*; *Roland et al., 2016*) and electrophysiology (*Sirotin et al., 2015*) studies. Interestingly, total spiking in piriform cortex decreased slightly at higher concentrations (*Figure 2C*). Nevertheless, a rate code could still be used to represent odor intensity if increased spiking in some cells was balanced by stronger suppression in others. Our data argue against this. First, the number of activated neurons did not change with concentration, although more cells were suppressed at higher concentrations (*Figure 2D*). Second, spiking in individual activated cells showed no systematic relationship to odorant concentration, although, again, suppression was weakly concentration-dependent (*Figure 2E*). Given the low firing-rate of piriform neurons, it seems unlikely that intensity could be encoded by weak suppression. Therefore, a simple rate code cannot be used to represent odor intensity in piriform cortex.

## Cortical synchrony increases with odor concentration

We therefore next asked if odor concentration might instead be encoded using temporal features of the population response. To reveal concentration-dependent changes in response dynamics we generated peri-stimulus time histograms (PSTHs, 10 ms kernel) for populations of simultaneously recorded piriform neurons. Population spiking from an example experiment is shown in *Figure 3A*. We observed a rapid increase in population spiking, peaking within 50–100 ms after inhalation. This was followed by a second phase of population spiking: at low concentrations the second phase was large and distinct from the first, however, as concentrations increased its latency decreased systematically so that the two phases of the response became more synchronous. Averaging across all experiments (*Figure 3B*), we again found a rapid increase in spiking that occurred within 50–100 ms after inhalation. This peak increased slightly at higher concentrations (0.03% mean ± st. dev.: 4.39 Hz ±1.33 Hz; 1%: 5.14 Hz ±2.16 Hz; repeated measures ANOVA, $F_{(3, 51)}$=3.78, p=0.0159, n = 18). This small increase in peak spiking, together with the small decrease in total spiking, indicates that there is a systematic, concentration-dependent redistribution of spike times (*Figure 3C*).

We observed sustained concentration-dependent suppression following the first peak. When averaging across cells in a population, suppression of some cells masks the activation of others. Therefore, rather than examining total population spike rate, we examined the timing of single cell responses. We selected cells with a detectable peak in the PSTH (i.e. cells that spiked at least once on any trial) in response to all four concentrations of either odorant (412/918 cell-odor pairs). Response times in some cells were similar across concentrations (*Figure 4A*, cells 1 and 2) while

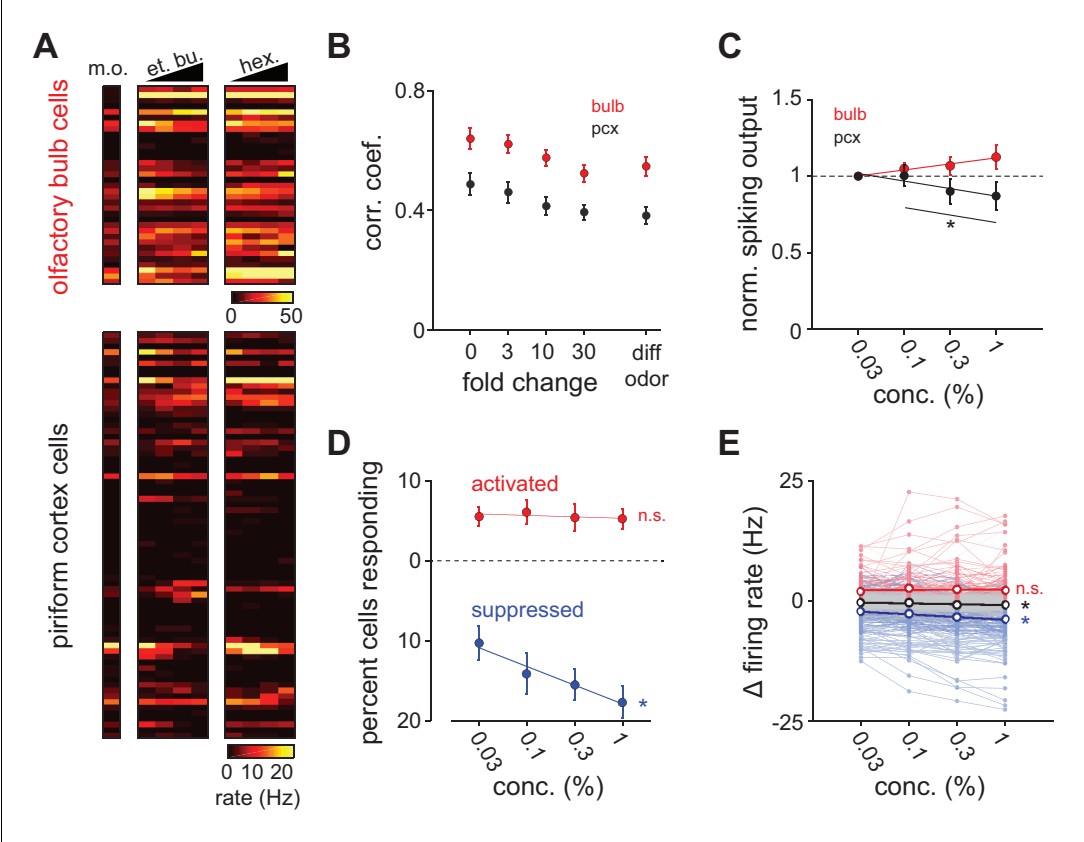

**Figure 2.** Concentration-dependent suppression in piriform cortex. (**A**) Simultaneous population recordings were obtained from olfactory bulb and ipsilateral piriform cortex in 6/9 experiments. Responses are represented as average firing rates over the first respiration cycle after odor delivery for one example experiment. (**B**) Average trial-by-trial correlation coefficients of population firing rate vectors in olfactory bulb (red circles; n = 6) and piriform cortex (black circles; n = 9), as a function of fold change in concentration for the identical odorant (0–30) or with a distinct odorant at any concentration (diff odor). (**C**) Total spiking output as a function of concentration in bulb (red circles; repeated measures ANOVA for concentrations 0.1–1%, $F_{(2, 22)}=1.66$, $p=0.21$) and piriform cortex (black circles; n = 12, repeated measures ANOVA for concentrations 0.1–1%, $F_{(2, 22)}=11.08$, $p<0.001$). Lines are fit to means as a function of concentration. Normalized spiking output did not differ significantly from 1 for any concentration in either region (one sample t-test, $p>0.05$). (**D**) Percent piriform cortex neurons significantly responding with increases in firing rate (red circles) or decreases in firing rate (blue circles; $p<0.05$, Wilcoxon rank-sum test vs. blank stimulus, n = 9). Lines are fit to means as a function of concentration. The percent of suppressed cells increased with concentration but the percent of activated cells did not change (repeated measures ANOVA, activated: $F_{(3, 24)}=0.25$, $p=0.86$; suppressed: $F_{(3, 24)}=10.83$, $p<0.001$). (**E**) Average piriform firing rates are not systematically concentration-dependent. Change in firing rate compared to blank responses for all cells for both concentration series (n = 918). Black unfilled circles are mean firing rate changes as a function of concentration for all recorded cells (s.e.m. smaller than marker size). Thin red and blue lines indicate concentration curves for cells that were significantly activated or suppressed for at least one concentration. Unfilled red and blue circles are mean changes in firing rate for these activated and suppressed cells, respectively. Thin gray lines are concentration curves for cells that were unresponsive at any concentration. Average firing rates changed for all responses and for suppressed responses as a function of concentration, but not for activated responses (repeated measures ANOVA, all: $F_{(3, 2751)}=23.00$, $p<0.001$; activated: $F_{(3, 354)}=1.24$, $p=0.30$; suppressed: $F_{(3, 705)}=40.08$, $p<0.001$).

The following figure supplement is available for figure 2:

**Figure supplement 1.** Reliable delivery of odor concentration series and concentration-dependent effects on breathing.

response latencies in other cells decreased systematically at higher concentrations (*Figure 4A*, cells 3 and 4). Note that although response latencies decreased, response amplitudes often remained constant or even decreased with concentration. Because most cells have low spontaneous firing rates (*Figure 1—figure supplement 1*) and brief response durations (*Figure 1—figure supplement 2*), we could define each cell's response by its latency to peak. Across the population, we found a systematic concentration-dependent decrease in response latency (*Figure 4B*).

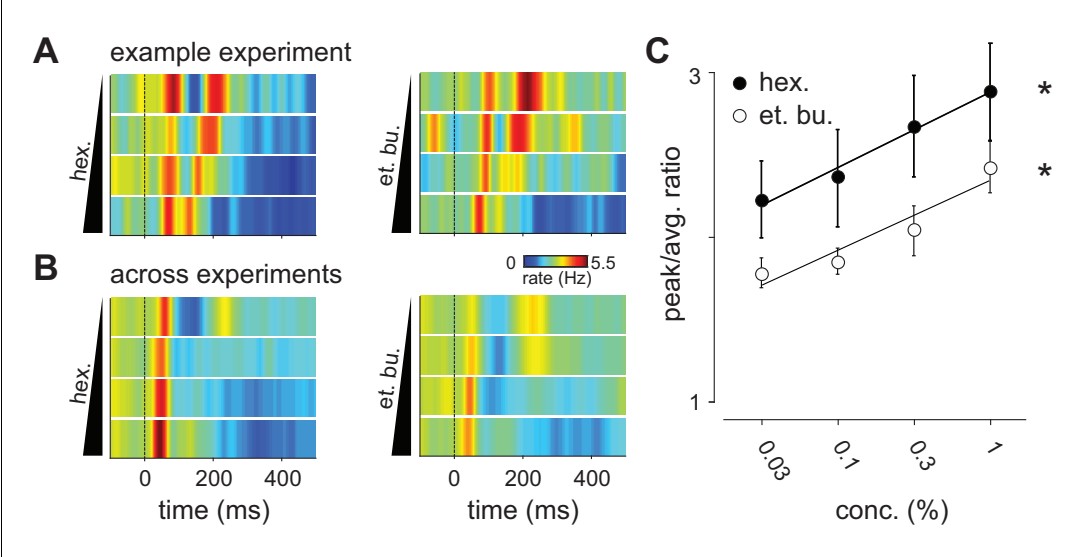

**Figure 3.** Distribution of spike times changes with concentration. (**A**) Population PSTHs averaged across, and normalized by, all 48 cells in an example experiment are shown as a function of concentration of ethyl butyrate (left) or hexanal (right). Dashed line indicates inhalation onset. (**B**) As above but with PSTHs averaged across all 459 recorded cells as a function of concentration. (**C**) The peak/average ratio, which indicates synchrony, increases significantly with concentration for both odors (repeated measures ANOVA, et. bu.: F(3, 24)=7.44, p<0.01, n = 9; hex.: F(3, 24)=6.29, p<0.01, n = 9).

To then determine how response times in individual cells change with concentration we sorted cells by their latencies to peak at a given concentration (0.3%, *Figure 4C*). This analysis revealed a systematic decrease in response latencies at higher concentrations (*Figure 4D*). However, the earliest responses were largely concentration-invariant, and concentration-dependent effects were greatest for cells that responded >100 ms later, suggesting again that there are functionally distinct classes of responses. In fact, the distributions of response latencies at all four concentrations were well fit by a mixture of Gaussian functions (*Figure 4E*, *Figure 4—figure supplement 1*). Time to peak of the first Gaussian barely changed across concentrations while the peak of the second Gaussian shifted systematically from 206 ms at 0.03% to 136 ms at 1% (*Figure 4F*). A third Gaussian was used to fit later components of the response, and parameters for this distribution were held constant while fitting the other two (*Figure 4—figure supplement 1*). Thus, the cortical odor response is composed of two phases that become more synchronous at higher concentrations.

## Different strategies for encoding odor identity and odor intensity

We next used a decoding analysis to reveal features of the cortical response that represent distinct features of the odor. To do this, we trained and tested a linear classifier on single-trial responses using three different types of response vectors that contained various amounts of information about the piriform response (*Figure 5A*). Binary response vectors simply indicate whether or not a cell responded (i.e. spike counts greater than one st. dev. above baseline) on each trial. By discarding all information about both the strength and timing of the response, as well as any information that may be conveyed by suppression, this approach determines how well a simple ensemble membership code can represent the odor. Spike count vectors contain information about the strength of each cell's response but no information about spike timing. Finally, spike count vectors subdivided into 30 ms bins to provide information about spike times.

We first asked the decoder to classify responses to six different odorants at a single concentration (0.3%). We varied the number of cells in different sized pseudopopulations, testing coding accuracy using 200 different, randomly selected subsets of cells per population size. Odorant classification accuracy, averaged across pseudopopulations, quickly rose using all three types of response vectors (mean number of cells to reach 90% accuracy: membership, 145 cells; summed, 115 cells; binned, 75 cells; *Figure 5B*). However, because individual cells convey different amounts of information about the stimulus, classifier outcomes varied across permutations. However, differences in classifier

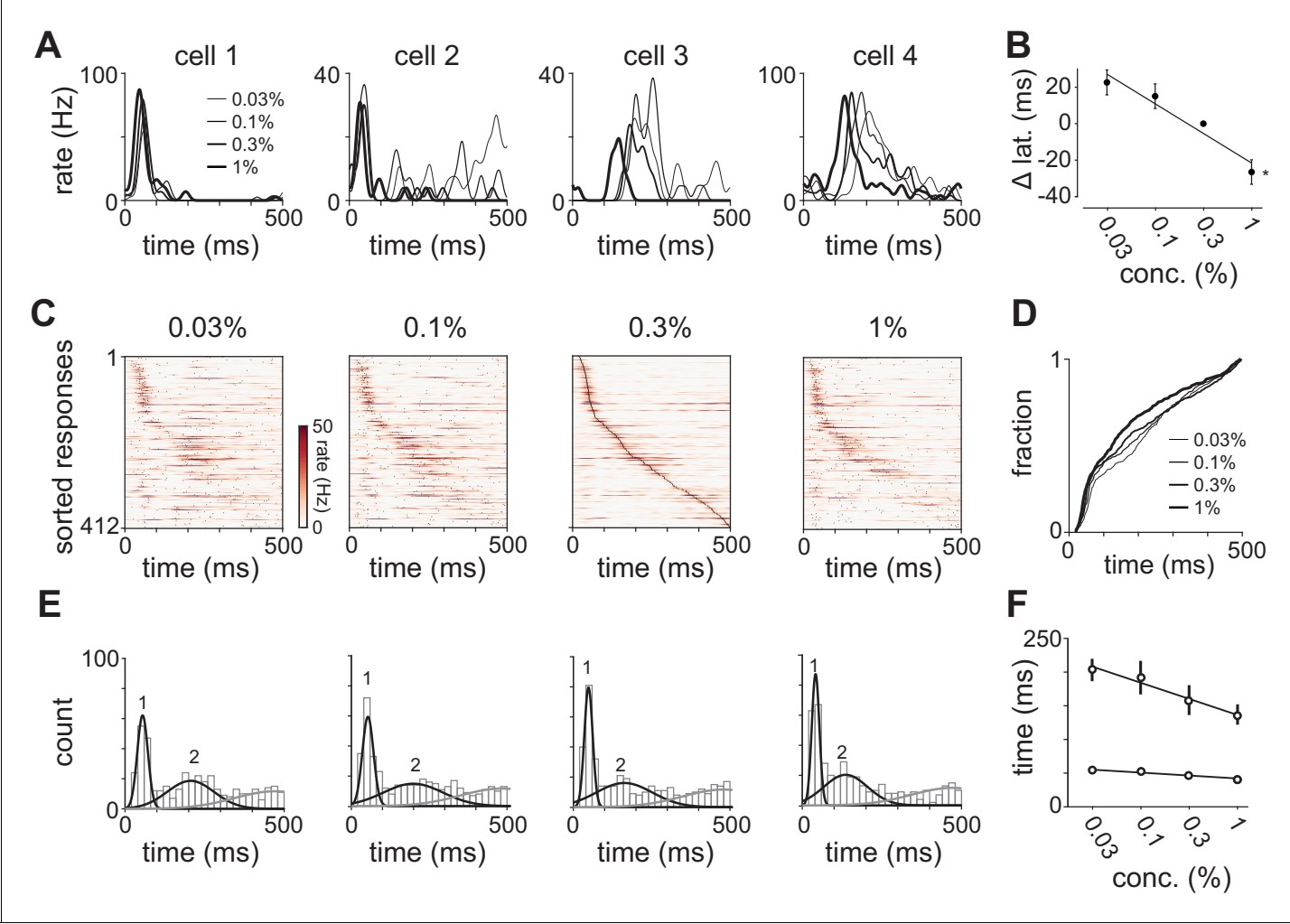

**Figure 4.** Odor concentration alters response timing. (A) Odor-evoked changes in firing rate for four cells in response to different odorant concentrations. Response latencies systematically decreased at higher concentrations in some activated cells but not in others. (B) Response latencies averaged across all piriform cortex cells decreased with concentration (repeated measures ANOVA for relative latencies at 0.03%, 0.1%, and 1%, F(2, 1834)=25.85, p<0.001; n = 918). Relative latencies at each concentration were significantly different from 0 (one sample t-test for 0.03%, 0.1%, and 1%, p<0.05, n = 918). Line is fit to relative latency means as a function of concentration. (C) PSTHs for responses in which a peak could be identified within the 500 ms response window. Cells are sorted by their latencies to peak at 0.3% v./v. with the same sorting order maintained for responses at different concentrations. Black dots indicate times of peak response. (D) Cumulative distributions of latencies to response peak at different concentrations. (E) Histograms of latency to peak distribution overlaid with fits for a mixture of three Gaussians. Note that while peak latencies decreased systematically at higher concentrations the timing of the earliest responses was largely concentration-invariant. (F) Timing of the first and second peaks of Gaussian fits (see *Figure 4—figure supplement 1*) for the distribution of latencies to peak activation (black circles, error bars are 2.5–97.5th percentile of bootstrap model fits).

The following figure supplement is available for figure 4:

**Figure supplement 1.** Fitting Gaussian mixture models to peak latency distributions.

performance with the same sets of cells using any of the three different coding schemes did not reach statistical significance over any tested population sizes (*Figure 5—figure supplement 2*). Odorant classification did not depend on stimulus concentration (*Figure 5—figure supplement 3*). Also, classifier performance using real populations of simultaneously recorded cells and pseudopopulations of cells sampled across experiments was equivalent (*Figure 5—figure supplement 4*).

We next asked the decoder to classify responses evoked by a given odorant at one of four different concentrations. In general, concentration classification was less accurate than classification of

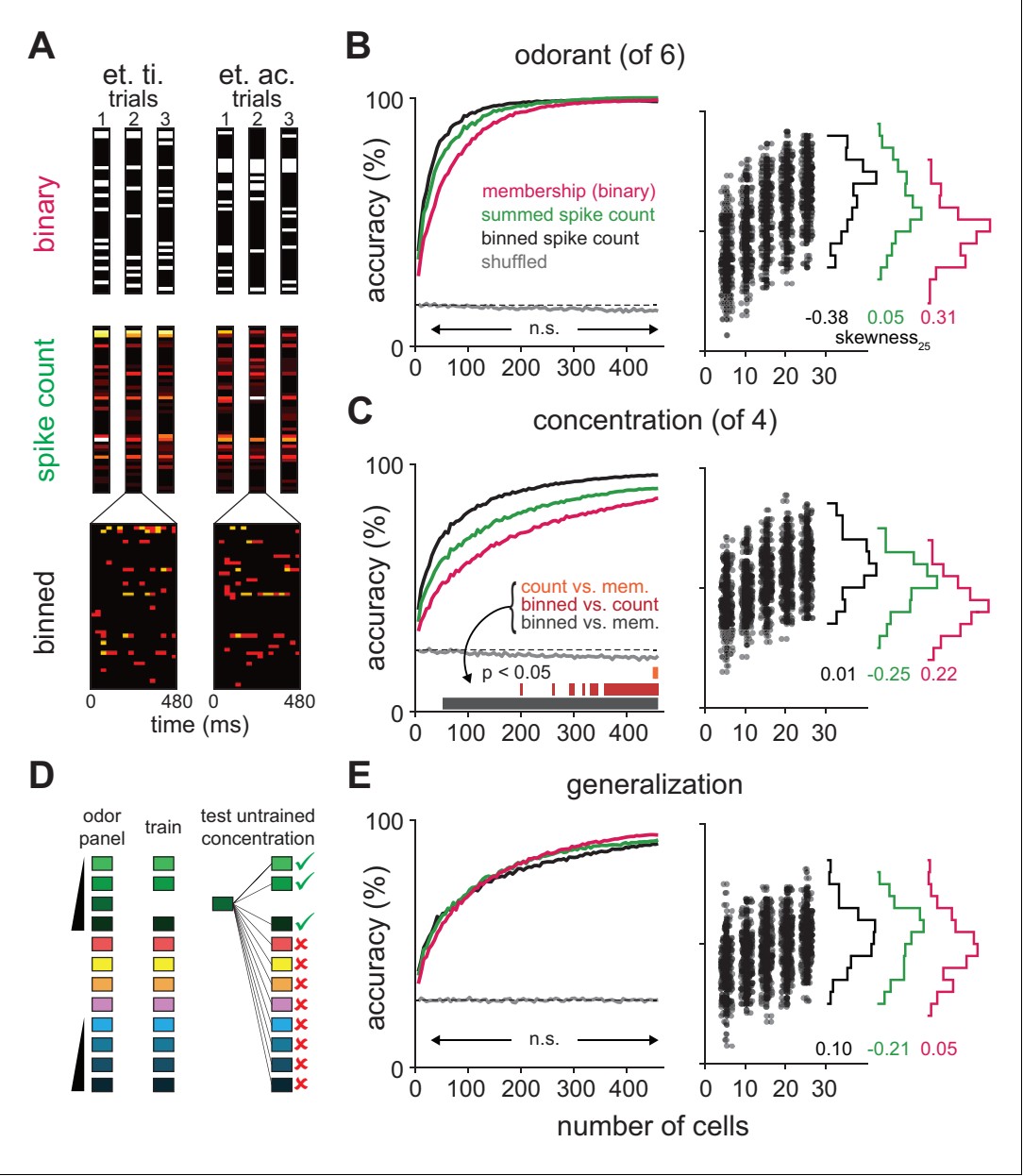

**Figure 5.** Coding strategies for representations of odor identity and intensity. (**A**) Feature vectors for classifier analysis were constructed as either spike counts summed over a 480 ms response window (middle), as binary activation vectors (top), or as spike counts for 16 concatenated 30 ms bins over the same time period (bottom). Binary activation vectors were constructed according to a 1 st. dev. threshold above mean spike counts on blank trials. (**B**) (Left) Classification accuracy for identifying one of six different odorants as a function of number of cells using binarized spike counts (magenta), summed spike counts (green) or binned spike counts (black). Data are mean accuracy for 200 permutations of pseudopopulation construction. Differences in classification accuracy using different coding schemes failed to reach statistical significance (p>0.05, see *Figure 5—figure supplement 2*). Classifier performance using shuffled trial label-spike counts (grey) was at chance levels (dashed line). (Right) Points illustrate accuracy for individual pseudopopulations at low sample sizes using binned spike count representations. Distribution of accuracies using different 25-cell populations is shown at the right for each coding scheme. (**C**) Same as B for classification for one of four concentrations of the same odorant. Spike time information markedly enhances intensity decoding accuracy. Accuracy for each permutation was the average of accuracy for the two different concentration series. Shading indicates points at which decoding accuracy was significantly different (p<0.05) for different coding schemes. (**D–E**) Odor identity classification accuracy for a classifier trained with all stimuli except for one concentration of the target odorant (n = 11 different stimuli) and then tested using

*Figure 5 continued on next page*

*Figure 5 continued*

the withheld responses. Accuracy was assessed for correctly identifying the odorant regardless of concentration. Differences in classification accuracy using different coding schemes failed to reach statistical significance (p>0.05).

The following figure supplements are available for figure 5:

**Figure supplement 1.** Limited heterogeneity of stimulus information among piriform cortex neurons.

**Figure supplement 2.** Significance testing of classifier performance.

**Figure supplement 3.** Odorant classification at different concentrations.

**Figure supplement 4.** Classification with real versus pseudopopulations.

**Figure supplement 5.** Optimal bin size for classification and alternate classifier method.

different odorants. Classifier performance using binary response vectors as inputs was relatively poor and did not reach the 90% accuracy criterion. However, classification was still much better than chance (*Figure 5C*), indicating that an odor at different concentrations activates highly overlapping but nevertheless distinct ensembles of piriform neurons. Classification accuracy only improved marginally when using spike count vectors (90% accuracy with 425 cells; *Figure 5—figure supplement 2B*). This result indicates that the strength of each cell's response provides little information about odor concentration and is consistent with data presented in *Figure 2*. However, if spike time information is used to represent odor intensity then using binned response vectors should improve classification. Indeed, classification accuracy and efficiency increased significantly using binned response vectors (90% accuracy with 210 cells; *Figure 5—figure supplement 2E*). Thus, we conclude that spike time information provides more information about odor concentration than spike count. Classifier performance did not qualitatively depend on bin size or type of classifier (*Figure 5—figure supplement 5*).

These analyses examine decoding of responses to different odorants or to different concentrations of a given odorant but they do not actually examine how odor identity per se is encoded. To do this, we derived a generalization task (*Figure 5D*). We used responses from our full odor stimulus panel in which some of the odorants were presented at multiple concentrations. We removed responses to an odorant at one concentration and trained the classifier on the remaining set of responses. We then asked the classifier to identify the odorant of the eliminated set of responses; that is, we asked the classifier to identify a familiar odor at a novel concentration. Classification accuracy was not statistically different using the three types of input vectors (*Figure 5E*) with, if anything, a trend towards slightly better performance using summed versus binned spike counts, and better performance still with a binary membership code (90% accuracy: binary, 310 cells; summed, 370 cells; binned, 425 cells; *Figure 5—figure supplement 2C,F,I*). In summary, we find that odor identity can be represented by the specific subsets of activated neurons, with no additional information provided by knowing when or how strongly each cell responded. Different concentrations of a given odor activate overlapping but nevertheless distinct cortical ensembles, and knowing when these ensembles respond provides substantially more information about odor concentration than knowing how strongly they respond.

## Are odor representations in piriform cortex sparse?

Odor representations are said to be sparse in piriform cortex (*Poo and Isaacson, 2009*; *Stettler and Axel, 2009*; *Miura et al., 2012*; *Otazu et al., 2015*) and analogous structures in invertebrates (*Perez-Orive et al., 2002*; *Honegger et al., 2011*; *Caron et al., 2013*). However, our measures of both lifetime sparseness (0.34 ± 0.21; n = 459 cells; *Figure 1—figure supplement 2*) and population sparseness (0.68 ± 0.10, n = 54 population-odors) in awake, head-fixed mice are considerably lower than in urethane-anesthetized rats (*Poo and Isaacson, 2009*) and slightly lower than in trained rats performing an odor-discrimination task (*Miura et al., 2012*). This led us to revisit the question of sparseness. The distribution of classifier performances with random subsets of cells

provides an alternate way to determine if cortical odor representations are sparse. Mouse piriform cortex contains ~$10^6$ neurons (*Srinivasan and Stevens, 2017*). If odor representations are sparse then randomly selecting small subsets of cells should produce a large and highly skewed distribution of classifier performances; in the vast majority of cases decoding accuracy will be poor as none of the selected cells will respond to the presented odors, but in a few cases decoding accuracy will be excellent if just the right combination of cells are selected. By contrast, if odor representations are dense then many cells will provide some information about most of the odors. In this case, accurate classification accuracy would be achieved using only a few cells, with a relatively small and normal distribution of classifier performances. Our data clearly support this second model. First, the range of performances is small. Randomly selected subpopulations of only 10 neurons had an average accuracy of >50%. Importantly, accuracy was above chance 98% of the time for this population size (*Figure 5B*). Second, we examined classifier performances when using subpopulations of 25 cells, when accuracy distributions are wide and should be highly skewed if odor representations are sparse. Instead, we found the distribution of performances were normally distributed when classifying different odorants (*Figure 5B*), concentrations (*Figure 5C*), or generalizing for odor identity (*Figure 5E*), using any of the decoding strategies. We therefore conclude that cortical odor representations are not especially sparse.

## Dissociating representations of identity and intensity

Our data suggest two different ways in which spike time information could be used to accurately represent odor intensity. First, we find a systematic concentration-dependent increase in the initial peak of the population PSTH (*Figure 3B*). Second, we find a small but systematic decrease in latencies of later-responding cells (*Figure 4D,F*). Either of these can explain why decoding concentration is better using binned versus summed response vectors. To compare the relative contributions of the early and late phases we asked when information becomes available for accurate classification of both odor identity and concentration. Using an expanding window as input (*Figure 6A*), we could accurately classify the odorant and generalize for odor identity within ~100 ms after inhalation (*Figure 6B*). However, intensity decoding accuracy remained poor during the initial ~100 ms and then increased steadily over the next ~100 ms. This result supports a model in which the earliest inputs generate a concentration-invariant representation of odor identity and odor intensity is encoded by the relative timing of later responses. The ability to generalize for identity should therefore deteriorate as responses become more concentration-dependent. To test this prediction we trained and tested the classifier using responses in a sliding 30 ms window as input (*Figure 6C*), which indicates what features are represented at different phases of the sniff. Odorant classification was accurate throughout the sniff, indicating that responses to the different odorants were always distinct. As predicted, identity generalization also increased rapidly while concentration decoding accuracy remained poor for the first ~100 ms. However, as concentration decoding improved, the ability to generalize across concentrations decreased (*Figure 6D*), consistent with a trade-off between a concentration-invariant representation of odor identity and a representation of odor intensity. Interestingly, trained mice take longer to discriminate differences in odor concentrations than they require to discriminate distinct odors (*Abraham et al., 2004*). The differences we find in the amount of time required to represent identity vs. intensity could underlie, at least in part, this difference in reaction times. We note, however, that strategies that our classifier was not designed to decode, in piriform or elsewhere, could support rapid decoding of odor concentration (*Resulaj and Rinberg, 2015*).

## Piriform cortex actively transforms olfactory bulb input

We next sought to examine the neural circuit mechanisms that underlie the concentration-dependence of cortical odor responses. If cortical responses simply reflect input from bulb then the two structures should have similar dynamics. However, only a few studies have examined how odor-response properties in populations of olfactory bulb mitral/tufted cells in awake rodents depend on odorant concentration (*Fukunaga et al., 2012*; *Otazu et al., 2015*; *Sirotin et al., 2015*). We therefore also examined odor responses recorded in olfactory bulb. As discussed above, total spike count did not change with concentration (*Figure 2*). As in piriform, the distribution of bulb response latencies was well fit with a mixture of Gaussians (*Figure 7* and *Figure 7—figure supplement 1*).

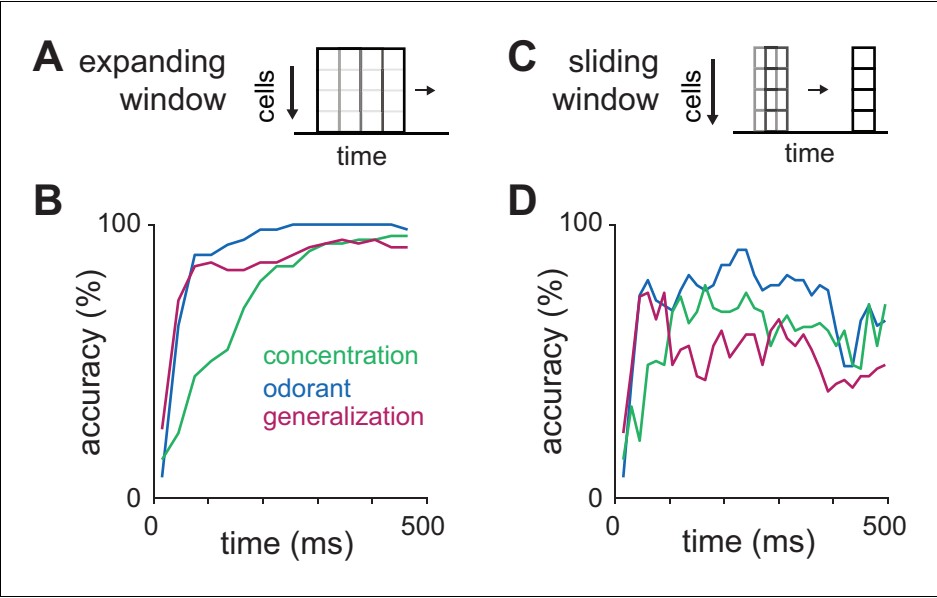

**Figure 6.** Dissociating identity and intensity coding. (A–B) Classification accuracy for odorant, intensity, or identity generalization as a function of integration time using an 'expanding window' consisting of feature vectors with increasing numbers of 30 ms bins for up to 480 ms. Pseudopopulations were constructed from all recorded cells (n = 459). The accuracy of intensity classification lags behind that for identity classification by ~100 ms and plateaus at a lower value. (C–D) Classification accuracy for odorant, intensity, or identity generalization as a function of time after inhalation using spike counts in a sliding 30 ms window as feature vectors. Note that while odorant classification accuracy remains elevated through the end of the sniff, generalization accuracy falls as concentration accuracy improves.

Latencies of the first responses were rapid and barely changed with concentration. These fast bulb responses overlapped with the first phase of piriform responses (**Figure 7D**), consistent with the earliest activated cells in bulb driving the fast responses in piriform. Again like piriform, latencies of the second peak in olfactory bulb responses decreased with concentration (**Figure 7—figure supplement 1D**). At low concentrations (0.03%) this peak appeared to occur earlier in bulb (192 ms) than piriform cortex (206 ms). However, interestingly, the second phase appeared to peak earlier in piriform (136 ms) than bulb (151 ms) at high concentrations (1%), although bootstrapped confidence intervals for the timing of these peaks overlapped.

This analysis suggests two ways in which odor information from olfactory bulb is actively transformed within piriform cortex. First, the second phase of the response appears more pronounced in bulb than cortex. To compare these directly, we overlaid the distribution of response latencies in bulb and cortex. At all concentrations, the early peak in the bulb response aligned with the early peak in the piriform response (**Figure 8A**). However, the second phase of the response was much more pronounced in bulb than piriform, indicating that later bulb input is actively suppressed within piriform cortex. This result suggests that although individual mitral/tufted cells may fire at different phases throughout the sniff cycle (**Bathellier et al., 2008**; **Cury and Uchida, 2010**; **Shusterman et al., 2011**) piriform cortex is preferentially responsive to the earliest-active bulb inputs.

Second, the shift in response latencies appears more pronounced in cortex than bulb. Variability from experiment-to-experiment may obscure some concentration-dependent changes in response dynamics. (For example, note the slightly longer latencies in the experiment shown in **Figure 3A** versus the average across experiments shown in **Figure 3B**). We therefore took advantage of our ability to record from large populations of neurons in bulb and piriform cortex to estimate concentration-dependent changes for each recording. We measured latency to first spike (**Figure 8B**) and latency to peak (**Figure 8C**) for each cell, and then fit a regression line through the population average to determine how the response changes with concentration. While latencies decreased systematically

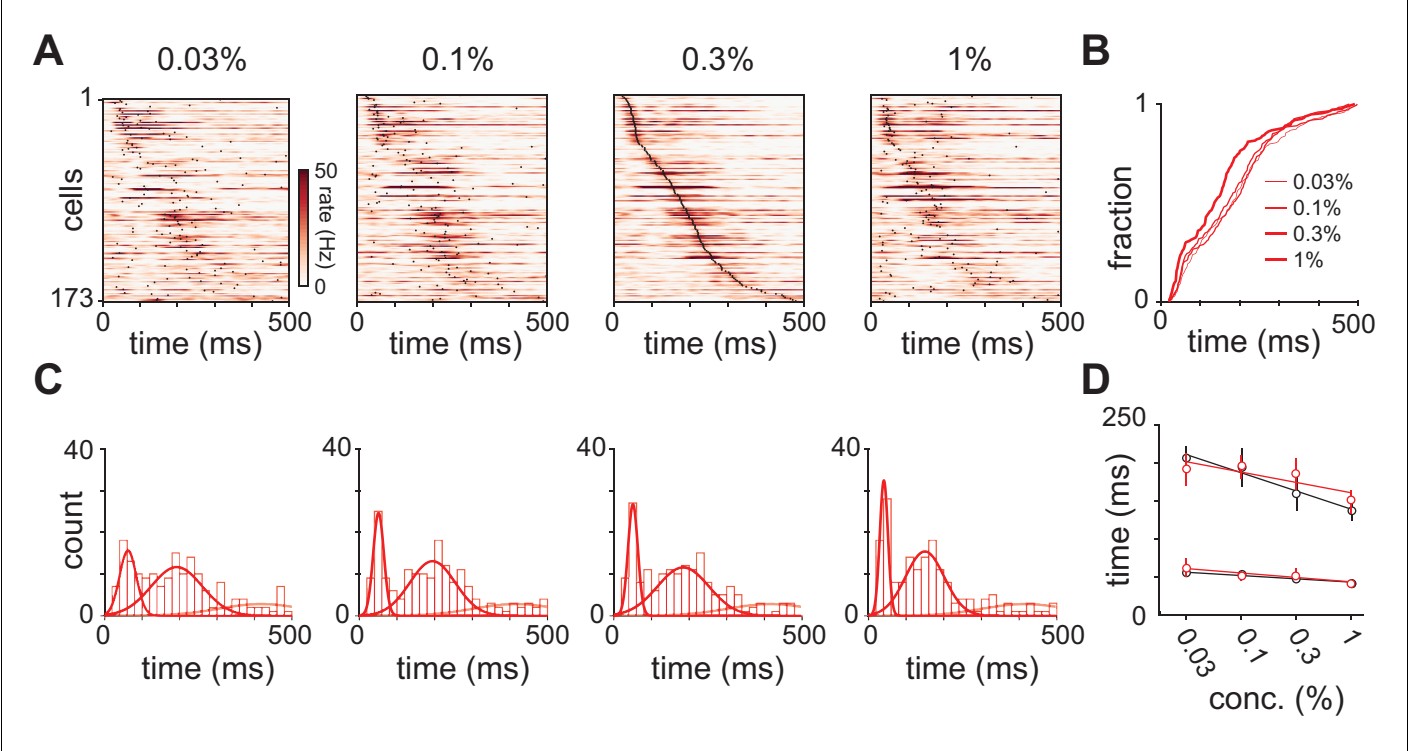

**Figure 7.** Response latencies for olfactory bulb cells change with concentration. (**A**) PSTHs for olfactory bulb responses sorted by their latencies to peak at 0.3% v./v. with the same sorting order maintained for responses at different concentrations. Black dots indicate time of peak responses. (**B**) Cumulative distributions of latencies to response peak at different concentrations. (**C**) Histograms of latencies to peak activation for olfactory bulb responses overlaid with fits for a mixture of three Gaussians. Olfactory bulb responses can clearly be divided into two phases but with different dynamics to those in piriform cortex. (**D**) Timing of the first and second peaks of Gaussian fits for the distribution of latencies to peak activation (red circles, error bars are 2.5–97.5th percentile of bootstrap model fits). Replotted results from *Figure 4F* show timing of peaks in piriform cortex (black circles).

The following figure supplement is available for figure 7:

**Figure supplement 1.** Fitting Gaussian mixture models to olfactory bulb peak latency distributions.

with concentration in both structures, both latency to first spike and latency to peak were more steeply concentration-dependent in piriform cortex (*Figure 8D,E*). Together, these data indicate an active transformation of bulb input within piriform cortex in which later inputs from bulb are suppressed and the population response is temporally sharpened with concentration.

## Intracortical inhibition is sharpened at higher odor concentrations

To probe the circuit mechanisms that mediate this transformation we obtained a separate set of recordings in mice that express channelrhodopsin-2 (ChR2+) in inhibitory interneurons (VGAT-hChR2-YFP; *Zhao et al., 2011*; *Hu et al., 2016*; *Large et al., 2016*) using an optic-fiber-coupled recording probe (*Figure 9A,B*). As before, probes were positioned to span layer II. We identified ChR2+ cells with a series of light pulses (1s) before and after presentation of the odor panel. While light pulses suppressed spontaneous spiking in most neurons, ~7% of cells (35/512, n = 9 recordings) exhibited robust and sustained light-evoked spiking (*Figure 9C*). We concluded that these cells were ChR2+ and defined them as inhibitory interneurons. Thus identified, inhibitory interneurons responded to most odors with an increase in spiking (lifetime sparseness: 0.11; *Figure 9D–F*). This broadly tuned activation of inhibitory cells is consistent with the widespread suppression we observe across the population (*Figure 1I*).

Identified inhibitory neurons had prolonged responses (*Figure 9D*). To determine the time course of inhibition in the cortex, we therefore averaged responses of all VGAT+ neurons to generate

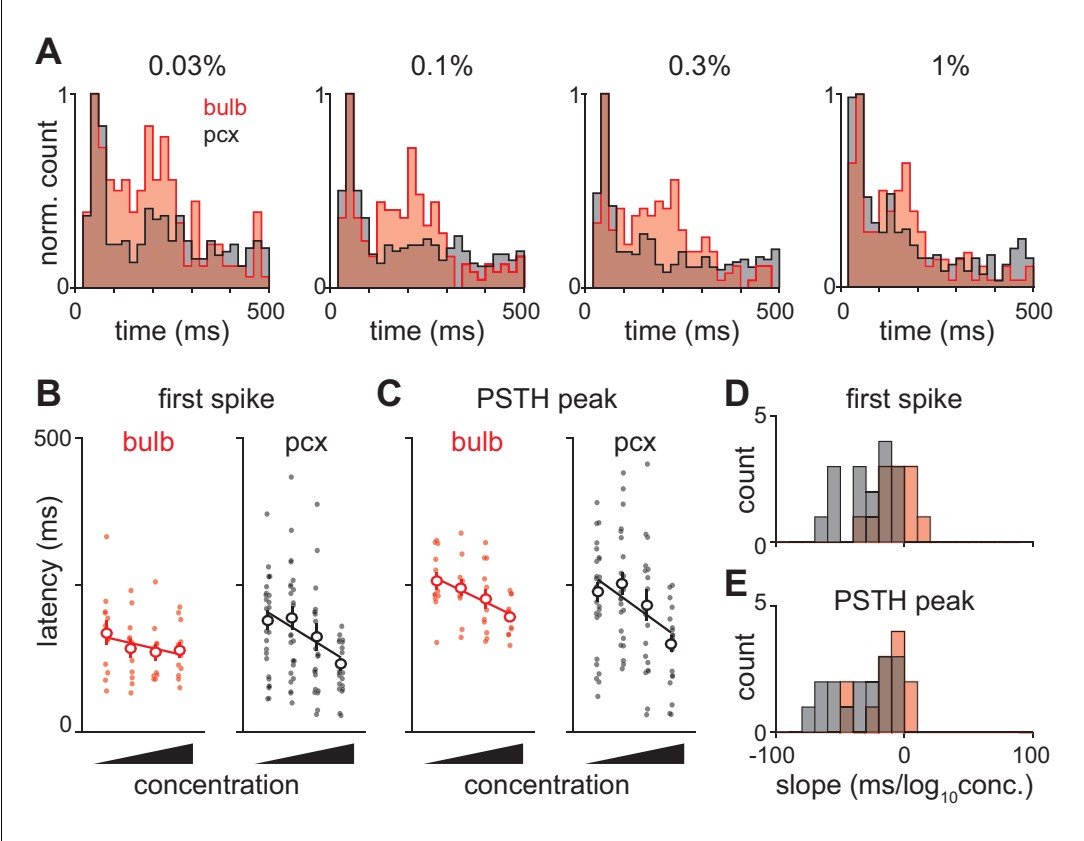

**Figure 8.** Olfactory bulb input is actively transformed in piriform cortex. (**A**) Response peak latency histograms normalized to peak value and overlaid for bulb and piriform. The second phase of responses is more prominent in bulb than in piriform cortex at all but the highest concentration (KS test: 0.03%, 0.0263; 0.1%, 0.0160; 0.3%, 0.0145; 1%, 0.1603). (**B–C**) Example experiment showing changes in ethyl butyrate response latencies in simultaneously recorded populations of bulb and piriform cells. Response latency for each cell was taken as either the average time to first spike (**B**) or the average time to PSTH peak (**C**). Points are average response latencies for each cell at different concentrations. Lines are fit to the average of these points. (**D–E**) Distribution of fitted slopes for bulb (n = 12, 6 experiments * 2 odors, red) and piriform (n = 16, 8 experiments * 2 odors, red). Fitted slopes were reliably steeper as a function of concentration for piriform populations (unpaired t-test: first-spike, p=0.004; PSTH peak, p=0.016).

inhibitory population PSTHs (*Figure 9G*). These revealed a sharp peak that rapidly decayed and then sustained at a lower level for the duration of the sniff. The peak amplitude of the PSTH increased systematically with concentration. Onset time was ~25 ms at all concentrations. However, latency to peak decreased (0.03%, 91 ms; 1%, 65 ms) and the initial peak became narrower (FWHM: 0.03%, 89 ms; 1%, 50 ms), indicating that cortical inhibition occurs earlier and sharpens at higher concentrations. This concentration dependence could arise from increased bulb input in the case of feedforward inhibition, or from the small increase in the initial peak of the population PSTH (*Figure 3*) in the case of feedback inhibition, or a combination of these. Future studies are required to dissociate the contribution of these two inhibitory processes.

Does this shift in inhibition play a role in controlling the dynamics of the cortical odor response? To address this question, we compared the inhibitory PSTHs with the timing of the two phases of the population response (*Figure 9H*). The first peak of the population response immediately preceded the increase in inhibition. Importantly, the second phase of the population response, whose timing changes with concentration, occurs immediately after the initial, sharp increase in inhibition has ended. This suggests that the onset of the second phase of the population response is shaped by intracortical inhibition.

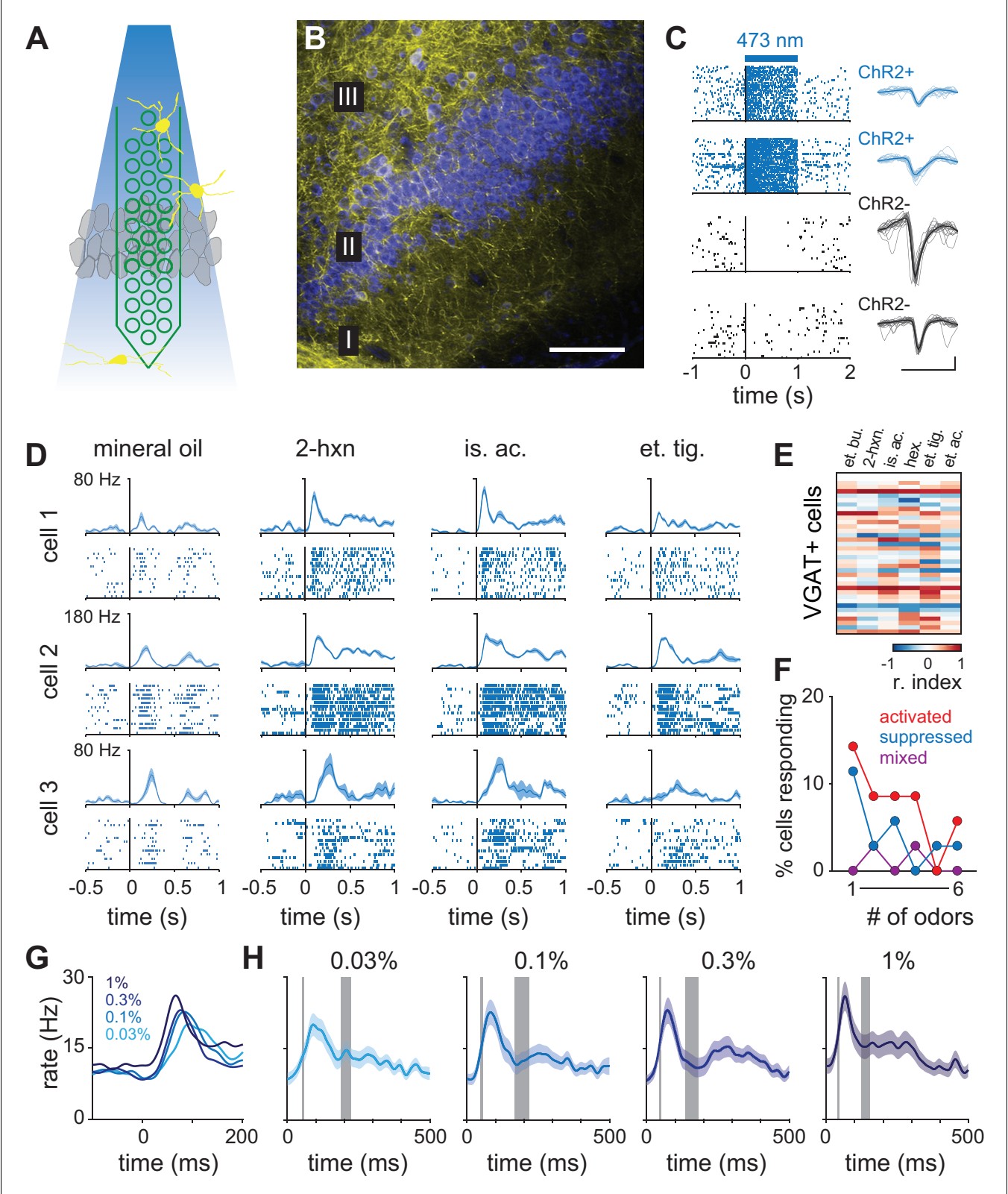

**Figure 9.** Dynamics of cortical inhibition indicates their role in shaping piriform responses. (A) Schematic of experiment. Optogenetic identification of ChR2-expressing neurons in piriform cortex of VGAT-ChR2-EYFP transgenic mice. (B) Coronal section of anterior piriform cortex of VGAT-ChR2-EYFP mouse showing dense innervation of layer II/III, consistent with feedback inhibition (yellow, anti-GFP; blue, neurotrace). Scale bar, 100 µm. (C) One second laser pulses (2.5 mW/mm2) evoke sustained responses in presumptive ChR2-expressing neurons (blue raster plots) but sustained suppression in

*Figure 9 continued*

most other cells (black). Scale bar, 50 µV, 1 ms. (D) PSTHs and raster plots for three VGAT+ cells responding to mineral oil control or three different odors. (E) Response matrix for all recorded VGAT+ cells (n = 35) responding to six different odors. (F) Tuning for all recorded VGAT+ cells. Responses tended to be positive and broadly-tuned. (G–H) Average PSTHs for all VGAT+ cell responses (lines and shading are mean ± s.e.m; n = 70, 35 cells x 2 odors) at four concentrations. (G) PSTHs shown on a short time scale. Stronger and earlier peaks occur at higher concentrations. (H) Average VGAT+ PSTHs overlaid with the bootstrap confidence intervals (gray bars) for the first and second peaks of the piriform cortex response latency distributions from *Figure 3*.

## Discussion

We proposed a multiplexed cortical odor-coding model in which odor identity is represented by specific ensembles of odor-responsive neurons while odor intensity is encoded using spike time information. In support of this model we find that odor identity can be decoded simply by knowing which cells respond (*Figure 4B,E*). Interestingly, information about spike count, spike time and odor-evoked suppression appear redundant for identity coding. By contrast, accurate and efficient odor intensity decoding requires information about spike times but is not improved by knowing spike counts (*Figure 4C*). Thus, knowing when a cell responds provides more information about odor intensity than knowing how strongly it responded. We note, however, that ensembles are not concentration-invariant, as concentration can be decoded reasonably well even when using a simple membership code. This indicates that intensity information is not exclusively temporal. Thus, the features used to represent identity and intensity are not fully independent. Nevertheless, we identify different and complementary coding strategies for representing these two stimulus features in piriform cortex.

### Bulb-to-piriform coding transformation

Different odors activate subsets of mitral cell responses with odorant-specific sequences (*Spors and Grinvald, 2002*; *Bathellier et al., 2008*; *Cury and Uchida, 2010*; *Shusterman et al., 2011*) and concentration-dependent latencies (*Cang and Isaacson, 2003*; *Margrie and Schaefer, 2003*; *Sirotin et al., 2015*). Our results are partially consistent with a prevailing model, initially proposed by Hopfield, for how odor representations are transformed and encoded in olfactory bulb, and could be decoded in piriform cortex (*Hopfield, 1995*; *Schaefer and Margrie, 2007*; *Junek et al., 2010*; *Schaefer and Margrie, 2012*; *Uchida et al., 2014*; *Wilson et al., 2015*). This model gives rise to several predictions: (1) The earliest-activated mitral cells will largely define the odor representation, and thus the ensemble of activated cortical cells, with later responses suppressed or 'discriminated against' (*Hopfield, 1995*); (2) Odor identity can be extracted from the sequence of the earliest-activated mitral cell responses and transformed into a representation that is robust to changes in concentration; (3) Spike time information should not inform representations of odor identity in cortex; (4) Odor intensity can be extracted from the latency to spike of the earliest-activated mitral cells. Consistent with these predictions, we find that piriform responds reliably to the early mitral cell responses but significant suppression within piriform cortex reduces the impact of mitral cells that spike later in the sniff (*Figure 8A*); that odor identity can be represented across concentrations without spike time information (*Figure 5E*; also *Miura et al., 2012*); and that spike time information is important for representing odor intensity (*Figure 4*, *Figure 5C*). However, if the earliest-activated mitral cells provide information about both odor identity and intensity then a relatively simple decoder should be able to classify identity and intensity simultaneously. However, we found that identity could be decoded nearly immediately after inhalation while accurate intensity decoding was restricted to the later phase of the response (*Figure 6*). This is consistent with our finding that concentration-dependent changes in response latencies were substantially more pronounced in the later phase of responses (*Figure 4D–F*). This dissociation of identity and intensity representations suggests that these may not be encoded as a single process upstream of the cortex and therefore requires some revision of the Hopfield model.

## Sparse coding in piriform cortex

Odor representations in piriform cortex have previously been described as 'sparse' or 'moderately sparse' (*Poo and Isaacson, 2009*; *Miura et al., 2012*). These conclusions were drawn from measurements of the odor selectivity of individual cells and the 'sparseness' metric originally developed by Rolls and Tovee (*Rolls and Tovee, 1995*). It remains true in our dataset that very few neurons responded statistically significantly to any stimulus, and our sparseness measurements are not substantially different from those previously reported for awake rodents (*Miura et al., 2012*). However, whether or not a neural code is sparse is supposed to reflect strategies for representing a large array of stimuli in a neural network. In a sparse code only a very small set of cells is sensitive to any given stimulus. Reducing overlap between stimuli increases representational capacity (*Olshausen and Field, 2004*). The key feature of a sparse code is therefore that most cells don't respond to any one stimulus. Here, we used our decoding analysis to specifically determine whether information about a given odor is encoded by a privileged set of neurons. We found that different features of the odor could always be decoded using small randomly selected pseudopopulations. These results do not support the conclusion that piriform representations are sparse.

## Representations of odor intensity

We find that temporal information is used for intensity decoding, and that some cells' response latencies decrease systematically at higher concentrations. Absolute response latencies could therefore be used to represent odor intensity using, for example, response times after inhalation. A general problem with latency codes is that a decoder must know when the stimulus started. However, the rapid concentration-invariant responses provide an internal time reference that obviates this problem in piriform cortex. A fast invariant and slower concentration-dependent response increases population synchrony at higher concentrations. Downstream neurons with distinct integration properties could therefore extract different features of the odor stimulus from this multiplexed representation (*Blumhagen et al., 2011*; *Haddad et al., 2013*). However, other strategies could also be used to represent identity and intensity. For example, *Roland et al. (2017)* found distinct subpopulations of neurons within piriform that have concentration-dependent or concentration-invariant responses.

## Mechanisms of temporal encoding

Multiple mechanisms, partially inherited from the bulb, could temporally distribute response latencies according to odorant concentration. First, glomerular response latencies in mirror-symmetric medial and lateral glomeruli become increasingly synchronous at higher odorant concentrations (*Zhou and Belluscio, 2012*). Second, although olfactory bulb mitral and tufted cells are typically conflated, these cells differ considerably in their spike thresholds, response latencies, and concentration dependence (*Nagayama et al., 2004*; *De Saint Jan et al., 2009*; *Gire et al., 2012*). We also find two distributions of response latencies in undifferentiated olfactory bulb recordings, again with a rapid and concentration-invariant peak and a later peak that shifts forward at higher concentrations (*Figure 7*). The fast and slow responses we see in piriform cortex could occur because individual anterior piriform cortex neurons receive converging inputs directly from both mitral and tufted cells, either directly (c.f. *Igarashi et al., 2012*), or through disynaptic routing through the anterior olfactory nucleus. In fact, the fast cortical response can help impose distinct phases in the bulb response via centrifugal projections (*Boyd et al., 2012*; *Kato et al., 2012*; *Markopoulos et al., 2012*; *Otazu et al., 2015*).

However, the concentration-dependent shift in response latencies in piriform cortex is not a simple reflection of biphasic bulb input. Concentration-dependent latencies change more steeply in piriform cortex than in olfactory bulb, indicating an active sharpening of cortical responses. Our data suggest this sharpening could be imposed by inhibition. Odors evoke an envelope of cortical inhibition that peaks earlier and is more narrow at higher concentrations. Direct, optogenetic activation of inhibitory interneurons suppresses both spontaneous (*Figure 9C*) and odor-evoked piriform spiking (not shown). Thus, a concentration-dependent sharpening of inhibition can gate the onset of the second, slower set of responses. Inhibition could be driven by direct input from olfactory bulb (*Luna and Schoppa, 2008*; *Poo and Isaacson, 2009*; *Stokes and Isaacson, 2010*; *Suzuki and Bekkers, 2012*), although feedback inhibitory interneurons may also contribute substantially to the inhibition envelope (*Franks et al., 2011*; *Suzuki and Bekkers, 2012*; *Large et al., 2016*).

## Other strategies to differentially represent odor features

Although piriform cells have often been treated as a homogenous population, there are at least three grossly different types of principal piriform neurons that differ in their position, morphology and connectivity (*Suzuki and Bekkers, 2011*). Different subtypes can be distinguished genetically (*Diodato et al., 2016*) or by their projection targets (*Chen et al., 2014*). We now identify multiple functionally distinct cell types. We find that some cells have almost no spontaneous activity and respond very selectively with rapid, reliable and brief bursts of spikes, similar to odor responses in the mushroom body, a homologous structure in insects (*Perez-Orive et al., 2002*). However, we also find over-represented subsets of broadly tuned cells. Segregation of these outputs could project different types of odor information to separate downstream targets. Furthermore, cells can either respond with activation or suppression, but individual cells are rarely activated by one odor and suppressed by another. This suggests the existence of distinct activated and suppressed cell classes. However, the finding that identity can be decoded with a binary, membership code excluding all information about suppression indicates this information is largely redundant, at least for identity coding. We find that spike count does not improve intensity or identity coding. This suggests that spike count may therefore represent other features of the stimulus, such as salience or valence (*Gire et al., 2013*).

Odor-driven behaviors require that an animal can extract different salient features of the odor stimulus from olfactory bulb output. We have identified complementary strategies giving rise to non-interfering representations of odor identity and odor intensity. A similar dissociation of different features of neural activity has been found in analogous structures. In hippocampus, for example, a cell's membership within a CA1 ensemble and its actual firing rate can differentiate between spatial and episodic representations (*Leutgeb et al., 2005*). Furthermore, multiplexed spike rate and spike time codes separately represent an animal's location and its speed traversing that location (*Huxter et al., 2003*). This dissociation of neural activity into membership, rate and time codes may thus be a general strategy for increasing bandwidth in cortical circuits.

## Materials and methods

All experimental protocols were approved by Duke University Institutional Animal Care and Use Committee. Unless stated otherwise all data are shown as mean ± s.e.m.

### Mice

Mice were adult (>P60, 20–24 g) offspring of *Emx*1-cre (+/+) breeding pairs obtained from The Jackson Laboratory (005628). Optical tagging experiments used VGAT-ChR2-EYFP mice (*Slc32a1-COP4\*H134R/EYFP*; 014548).

### Headpost implantation procedure

Mice were anesthetized with ketamine (50 mg/kg)/xylazine (2 mg/kg) and maintained on isoflurane anesthesia (1–1.5% in O2) during headpost implantation. An incision was made on the midline and the periosteum was removed from the entire skull surface. Recording penetration sites were marked on the skull and a custom titanium headpost was lowered into place with a stereotaxic arm and fixed to the skull with Metabond (Parkell, Inc.). Animals received buprenorphine 0.1 mg/kg s.c. at the end of the surgical procedure and recovered 2–4 days before head-fixed procedures.

### Head-fixation

Mice were habituated to head-fixation and tube restraint for 15–30 min on each of the two days prior to experiments. The head post was held in place by two clamps attached to ThorLabs (Newton, NJ) posts. A hinged 50 ml Falcon tube on top of a heating pad (FHC, Bowdoin, ME) supported and restrained the body allowing maintenance of body temperature during anesthesia in the head-fixed apparatus.

### Data acquisition

Electrophysiological signals were acquired with a 32-site polytrode acute probe (A1 × 32-Poly3-5mm-25s-177, Neuronexus, Ann Arbor, MI) through an A32-OM32 adaptor (Neuronexus) connected

to a Cereplex digital headstage (Blackrock Microsystems, Salt Lake City UT). Unfiltered signals were digitized at 30 kHz at the headstage and recorded by a Cerebus multichannel data acquisition system (BlackRock Microsystems). Experimental events and respiration signal were acquired at 2 kHz by analog inputs of the Cerebus system. Respiration was monitored with a microbridge mass airflow sensor (Honeywell AWM3300V, Morris Plains, NJ) positioned directly opposite the animal's nose. Negative airflow corresponds to inhalation and negative changes in the voltage of the sensor output.

## Electrode placement

The recording probe was positioned in the anterior piriform cortex using a Patchstar Micromanipulator (Scientifica, UK). Coordinates marked on the skull were used to align the skull in the head-fixed apparatus for stereotaxic targeting. On the day of recording, mice were lightly anesthetized with isoflurane (1% in $O_2$), a craniotomy was opened over the recording site, the dura mater was removed, and the probe was positioned at 1.32 mm anterior and 3.8 mm lateral from bregma. Recordings were targeted 3.5–4 mm ventral from the brain surface at this position with adjustment according to the local field potential (LFP) and spiking activity monitored online. Electrode sites on the polytrode span 275 µm along the dorsal-ventral axis. The probe was lowered until a band of intense spiking activity covering 30–40% of electrode sites near the correct ventral coordinate was observed, reflecting the densely packed layer II of piriform cortex. Recording sites ventral to the layer were nearly silent and, under light anesthesia, LFP signals across all electrode sites often showed some synchronization with respiration. Prior to recording, the back of the probe was painted with a fluorescent dye (diI, Life Technologies) and probe position was confirmed post hoc in histological sections.

For simultaneous olfactory bulb recordings, a craniotomy was opened over the ipsilateral olfactory bulb. A micromanipulator holding the recording probe was set to a 10-degree angle in the coronal plane, targeting the ventrolateral mitral cell layer. The probe was initially positioned above the center of the olfactory bulb and then lowered along this angle through the dorsal mitral cell and granule layers until encountering a dense band of high-frequency activity signifying the targeted mitral cell layer, typically between 1.2 and 1.8 mm from the bulb surface.

## Spike sorting and waveform characteristics

Individual piriform units were isolated using the Klusta Suite (https://github.com/kwikteam). Spikes were identified by Spikedetekt when the high-pass filtered voltage trace crosses threshold on adjacent electrode sites. Spikes were automatically sorted into clusters with a 'Masked Expectation-Maximization' algorithm (*Kadir et al., 2014*), and the identified clusters are manually adjusted using the Phy interface (*Rossant et al., 2016*). Clusters with large numbers of refractory period violations (<2 ms) in the autocorrelogram were removed from the dataset. Pairs of units with similar waveforms and coordinated refractory periods in the cross-correlogram were combined into single clusters. Isolation distance (*Schmitzer-Torbert et al., 2005*) was calculated using the IsolationDistance function from the FMAToolbox (http://fmatoolbox.sourceforge.net). To allow comparison with previous studies that used the first three principal components of waveform from tetrode recordings, isolation distance was computed using combinations of 12 principal component-based waveform features derived from a subset of the 32 polytrode channels. Extracellular waveform features were characterized according to standard measures: half-width, trough-to-peak time, trough-to-peak amplitude, and peak amplitude asymmetry (*Barthó et al., 2004*). Unit position with respect to electrode sites was characterized as the average of all electrode site positions weighted by the wave amplitude on each electrode.

## Spontaneous firing rate and phase distributions

Spontaneous firing rates were assessed during inter-trial time segments greater than 4 s after stimulus offset. To determine phase preferences of spontaneous activity each spike was assigned a phase in the respiration cycle by linear interpolation between 0 (the start of inhalation) and $\pi$ (the start of exhalation). Phase histograms (10° bins) for each isolated unit were then constructed from spontaneous spiking and sorted according to peak phase.

### Respiration alignment

Respiration traces sampled at 2 kHz were smoothed with a second-order Savitzky-Golay filter in 200 ms frames and locally detrended in 1.5 s windows with 1 s overlap. The start of inhalation and exhalation were defined as zero-crossings before and after large negative peaks in the smoothed, detrended signal, respectively.

### Odor stimuli

The odor stimulus set consisted of hexanal (Aldrich 115606), ethyl butyrate (Aldrich E15701), ethyl acetate (Sigma-Aldrich [St. Louis, MO], 34858), 2-hexanone (Fluka [Mexico], 02473), isoamyl acetate (Tokyo Chemical Industry [Cambridge, MA], A0033), and ethyl tiglate (Alfa Aesar [Haverhill, MA], A12029). All odorants were diluted to 0.6% v./v. in mineral oil and further air-diluted to 0.3% during odor delivery. To examine intensity coding, additional dilutions of 0.06%, 0.2%, and 2% were prepared for ethyl butyrate and hexanal, creating two concentration series with final dilutions of 0.03%, 0.1%, 0.3% and 1%% v./v. The complete odor panel therefore consisted of 12 different odor stimuli.

### Odor delivery

Odors were delivered using a custom olfactometer. Charcoal filtered air was routed into three flow-meters, one of which carried a neutral air stream at 1 LPM and the other two of which carried 0.5 LPM each to two eight-valve banks. Air passing through one of the valve banks continued to one of eight 20 mL amber vials containing 5 mL of odorant solution or pure mineral oil, then through a Teflon, eight-inlet manifold, and then the two separate valve bank streams were recombined to produce a 1 LPM odorized air stream. One valve containing mineral oil on either bank was always open and diluted the output of the opposite bank by half. The neutral and odorized airstreams were routed to a three-way isolation 'final' valve (NResearch PN: SH360T042, West Caldwell, NJ) situated near the head-fixed apparatus. Normally, the neutral stream was routed to the nose and the odor stream to a vacuum exhaust. Unless stated otherwise all odorants were delivered at a final concentration of 0.3%. In between trials the odor stream consisted of air passing through two 'blank' vials containing only mineral oil.

A custom MATLAB script controlled the opening and closing of valves in valve banks and the switching of the final valve. During an odor presentation, the odor valve was selected at the valve bank and the odor stream flowing to exhaust became equilibrated for one-second while respiration was sampled to determine the mean of the respiration signal. Final valve switching was triggered on the beginning of exhalation, indicated by a positive-going mean crossing of the respiration signal following a negative peak. Final valve switching simultaneously rerouted the odor stream to the animal's nose and the neutral stream to exhaust, and switched back after 1 s. The olfactometer was regularly calibrated such that final valve switching produced minimal changes in the signal from a flow sensor situated at the nose outlet of the final valve. Consistent odor delivery was confirmed with a photoionization detector (PID) reading from the exhaust. Each stimulus was presented 10–15 times in a given experimental condition. In pilot experiments we presented odors over a range of inter trial intervals and found responses were stable when presented every 5 s or more (not shown). Odors were presented every 10 s.

### Individual cell-odor responses

We computed smoothed kernel density functions (KDF) with a 10 ms Gaussian kernel (using the psth routine from the Chronux toolbox: www.chronux.org) to visualize trial-averaged firing rates as a function of time from inhalation onset and to define response latencies for each cell-odor pair. To define peak latencies, KDFs were computed from spike times occurring with a 500 ms response window following inhalation. Peak latency was the time of the maximum of this KDF. In cases where the KDF maximum was at the window edge, indicating falling from or rising to a peak outside of the response window, peak latency was undefined. Response duration was the width at 25 or 50% of the maximum of this peak. The reliability of differences between responses to odor and mineral oil in a given time window or breath cycle were quantified by computing the area under the receiver operating characteristic curve (auROC) using the distributions of spike counts under the two stimulus conditions. The response index was obtained by multiplying the auROC by 2 and subtracting 1. A response index of −1 reflects unequivocal suppression, 0 reflects no change, and 1 reflects

unambiguous activation of responses versus blank. We identified responses that were significantly different between odor and mineral oil using a Wilcoxon rank-sum test with these same spike count distributions.

## Population response vectors and total spiking output

The pattern of spiking activity in response to odor across all neurons within a time window is represented as a population spike count vector. To identify patterns in the overall spiking activity of simultaneously recorded neurons we also define total spiking output as the re-aggregation of spike trains of all sorted units integrated over the first sniff.

## Tuning and sparseness

Tuning selectivity was characterized by the percent of cells significantly responding to increasing numbers of odors. To compare the observed tuning curves to the expectation of randomly assigned responses, we shuffled cell identities within an odor 100 times (see *Figure 1—figure supplement 3*), preserving the total response distribution for each stimulus, and recomputed tuning selectivity using these shuffled population responses.

Lifetime sparseness is a measure of the non-uniformity, or peakedness, of the distribution of responses to different odors for a given cell, whereas population sparseness measures the same for the distribution of cell responses for a given odor (*Rolls and Tovee, 1995*). Lifetime sparseness ($S_L$) for each cell was calculated as in (*Vinje and Gallant, 2000*):

$$S_L = \{1 - [(\Sigma r_i/n)^2/\Sigma(r_i^2/n)]\}/[1 - (1/n)]$$

where $r_i$ is the trial-averaged response to the $i$th odor and $n$ is the number of odors. Population sparseness ($S_P$) for each odor was calculated with the same formula, with $r_i$ as the trial-averaged response of the $i$th cell (averaged across trials) and $n$ as the number of cells.

## Trial-trial population vector correlations

The similarity of population odor responses, defined as spike count vectors within the first sniff, was quantified using Spearman's rank correlation coefficient to account for the non-normal distribution of spike counts across the population. Qualitatively similar, but higher, correlations were obtained using Pearson's correlation coefficient. Within an experiment, correlations for all trial-pairs were calculated (i.e. 10 trials for two odors = 100 correlations). The correlation between two stimuli or between a stimulus and itself was then taken as the average of these correlations.

## Classifiers

Linear classifiers were implemented using custom MATLAB scripts and the Statistics and Machine Learning Toolbox. Odor classification accuracy based on population responses was measured using a Euclidean distance classifier with Leave-One-Out cross-validation (*Campbell et al., 2013*). Mean population responses were computed for all odors across trials, excluding one trial. The excluded trial was then classified to the odor with the mean population response with the minimum Euclidean distance from the trial population response. This process was repeated for all trials of all odors. Accuracy was computed as the average percent of correct classifications across odor categories. Results were qualitatively similar using a support vector machine with a linear kernel (Error-correcting output codes multiclass model, MATLAB Statistics and Machine Learning Toolbox).

## Classification tasks

Decoding of different features of the odor stimulus from neural activity was assessed using three different classification tasks. First, for *odorant classification*, responses to six distinct monomolecular odorants presented at 0.3% were used as the training and testing data. Second, for *concentration classification*, responses to a single odorant at four concentrations, ranging from 0.03–1% were used. Our stimulus panel contained two odorants presented at multiple concentrations. The classifier was trained and tested separately for each odorant and the concentration classification accuracy was averaged for the two odorants. Third, for *odor identity generalization*, we excluded all trials for a given stimulus concentration from the training set, so that the training set consisted of 11 stimuli, three of which were different dilutions of the same identity as the excluded stimulus. Accuracy in this

task was then measured by presenting the classifier with trials of the excluded stimulus and scoring as correct any predictions that match the odor identity regardless of concentration (chance accuracy = 3/11). This was repeated for each stimulus belonging to a concentration series (8 total) and accuracy was averaged across these stimuli.

## Classification features

The feature vectors for *spike count* classification were the spike counts for each cell during the 480 ms following inhalation. For *binned* classification, trial PSTHs for each cell were computed with 30 ms bins up to 480 ms after inhalation onset, and then concatenated to form a feature vector. For *binary* feature vectors, a threshold of mean +1 st. dev. of the response on blank trials was set for each cell, and spike counts for each trial were recoded as responding (1) or not (0) based on comparison to this threshold. To assess the effect of population size on classification accuracy, randomly selected cells from our entire recorded data set were combined to form a pseudo-population of a given size. For each population size, the random selection and classification was repeated 200 times, and the results were averaged. Decoding analyses of the temporal evolution of odor representations used pseudo-population vectors assembled from all recorded cells. Classification was performed as described with feature vectors that consisted of either an expanding window of increasing numbers of 30 ms bins or in 30 ms windows at increasing times after inhalation up to 480 ms.

Classification was also performed on shuffled data in which the trial labels were randomly assigned to new odor categories. Repeating this shuffling procedure 200 times and averaging the results produced accuracy indistinguishable from the theoretical chance level of accuracy, (# of stimuli) $^{-1}$.

## Fitting Gaussian mixture distributions

Response latencies were taken as the peak of the trial-average kernel density function (computed with a at 10 ms Gaussian kernel) for each cell-odor pair. Latencies were included in this analysis if peak was found between 0 and 0.5 s for each concentration for a given cell-odor pair. For this analysis response latencies were combined across all recording before attempting to fit. The distributions of response latencies in olfactory bulb and piriform cortex were fit separate at each concentration with a mixture of Gaussian distributions (each of which was truncated between 0 and 0.5 s) using the maximum likelihood estimation (mle) function in the Statistics and Machine Learning Toolbox in MATLAB. The model was initialized with cluster assignments obtained using k-means clustering (k = number of mixture components). For each fit, the algorithm was allowed to run to convergence of until completion of 8000 iterations or function evaluations. Fits that did not converge were labeled and removed from further analysis. For each set of latencies and parameters, the algorithm was reinitialized five times.

We obtained confidence intervals for parameter estimates by creating 1000 bootstrap samples for each set of latencies, sampling with replacement to create an equivalent sized surrogate sample. The fitting procedure was repeated as above and confidence intervals were defined as the 2.5–97.5th percentile for each parameter estimate.

For certain analyses, the distributions of latencies were re-fit while some parameters were held constant and others were allowed to vary. In these cases, the fixed parameters were held at the mean of parameter estimates across all concentrations from the original fitting procedure.

## Fitting concentration-dependence of response latencies

We measured the concentration-dependence of mean response latencies for each simultaneously recorded population and each odor as the slope of a regression line fit to the mean of response latencies within a population-odor pair across concentrations. To allow analysis of first spike latency, latencies in this analysis were computed on a trial-by-trial basis. For each cell-concentration pair, we measured both the time to the first spike after inhalation for each trial and the time to peak of the trial's kernel density function and then averaged these values across trials. For either measure, responses were included if latencies could be measured on more than one trial. Concentration-specific latencies were then averaged across cells and a regression line was fit to obtain a concentration-latency slope. For one piriform population, no latencies could be retrieved for either odor, leaving eight piriform recordings and six olfactory bulb recordings for statistical comparison.

## Acknowledgements

We thank Nao Uchida for help with respiration alignment; Mark Dalgetty for technical assistance; Greg Field, Andrew Fink, Lindsey Glickfeld, Benjamin Roland and Carl Schoonover for helpful discussions; Court Hull, Steve Lisberger, Andreas Schaefer, and Fan Wang for reading early versions of the manuscript; and Alexander Fleischmann for extensive collaborations. This work was supported by grants from NIDCD (DC009839 and DC015525) and the Edward Mallinckrodt Jr. Foundation.

## Additional information

### Funding

| Funder | Grant reference number | Author |
| --- | --- | --- |
| National Institutes of Health | DC009839 | Kevin M Franks |
| National Institutes of Health | DC015525 | Kevin M Franks |
| Edward Mallinckrodt Jr. Foundation | | Kevin M Franks |

The funders had no role in study design, data collection and interpretation, or the decision to submit the work for publication.

### Author contributions

KAB, Conceptualization, Data curation, Software, Formal analysis, Validation, Investigation, Visualization, Methodology, Writing—original draft, Writing—review and editing; KMF, Conceptualization, Resources, Data curation, Formal analysis, Supervision, Funding acquisition, Validation, Investigation, Visualization, Methodology, Writing—original draft, Project administration, Writing—review and editing

### Author ORCIDs

Kevin A Bolding, http://orcid.org/0000-0002-2271-5280
Kevin M Franks, http://orcid.org/0000-0002-6386-9518

### Ethics

Animal experimentation: All experimental protocols were approved by Duke University Institutional Animal Care and Use Committee according to protocols A243-12-09 and A220-15-08.

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
