## [Decision Letter]

Thank you for submitting your article "Complementary codes for odor identity and intensity in olfactory cortex" for consideration by *eLife*. Your article has been favorably evaluated by a Senior Editor and three reviewers, one of whom is a member of our Board of Reviewing Editors. The following individual involved in review of your submission has agreed to reveal his identity: Dmitry Rinberg (Reviewer #3).

The reviewers have discussed the reviews with one another and the Reviewing Editor has drafted this decision to help you prepare a revised submission.

Summary:

The authors record ensembles of individual neurons in the anterior piriform cortex of awake head-fixed rats using polytrode probes, while changing identity and concentration of different odorants. They investigate possible coding strategies for odor identity and intensity, and propose that two different codes – a rate code (for identity) and a time code (for intensity) are used in a multiplexed fashion to extract these stimulus features. All reviewers agreed that the study was potentially a valuable advance in our understanding of odor and concentration representations in the piriform cortex. They felt that the data analysis and presentation could be strengthened.

Essential revisions:

1) The authors should analyze the concentration coding using a formal model that quantifies fit and confidence. This should address the observations of latency, inhibitory contributions, and synchrony.

2) The classifier analysis should be should be extended to establish whether the coding is sparse.

3) The authors should revisit the temporal analysis and conclusions, and clearly state how strong is the confidence in them.

4) Some details of methodology need further elaboration: e.g., analysis, bulb recordings, and unit isolation.

The outcome of the analysis above may suggest that the claims in the title should be reconsidered.

Additional specific comments are also indicated in individual reviewer comments below.

*Reviewer #1:*

This is a systematic and large study of single-unit activity in anterior piriform cortex of awake mice. The authors examine odor response patterns for monomolecular odorants. This is a valuable dataset as it covers many neurons, in awake animals, and over several odors. Consistent with previous data, the authors find a spectrum of response strengths, precision, and selectivity.

The authors have a key result on concentration-dependent coding. This is valuable but could be sharpened. There is quite a bit of data here, which is potentially very informative, however the analysis could probably be taken a bit further.

Then the authors pursue an analysis of concentration-dependent latency. There are two aspects to this, both of which could be strengthened.

1) First, can the concentration-dependence be explained simply by bulbar concentration-dependent latency? To their credit, the authors have directly recorded and shown the bulbar concentration-dependence. At present it isn't clear if the piriform is doing any substantial transformation of the response.

2) There is a nice experiment wherein the authors investigate the mechanisms behind concentration coding by optogenetically targeting inhibitory interneurons and delivering light pulses to identify the neurons from the recorded dataset. They characterized the responses of these neurons. Using this, the authors seem to have an implicit model whereby the inhibitory timing in the piriform acts to separate out the second peak in excitatory neurons from the first one. This implies that concentration dependence of the inhibitory peaks (Figure 6) should account for the shift in second peak timing for the remaining cells (Figure 6, Figure 3). Does such a model work? Does it improve on the already present concentration-dependent latency response from the bulb? Whether or not my interpretation is what the authors had in mind, it would be useful for the authors to make their response model explicit, and test it.

*Reviewer #2:*

Bolding & Franks in 'Complementary codes for odor identity and intensity in olfactory cortex' record ensembles of individual neurons in the anterior piriform cortex of awake head-fixed rats using polytrode probes, while changing identity and concentration of different odorants. They investigate possible coding strategies for odor identity and intensity, and propose that two different codes – a rate code (for identity) and a time code (for intensity) are used in a multiplexed fashion to extract these stimulus features. Consistent with previous reports, the authors find sparse odor representations in the cortex. Individual responsive neurons were unambiguously suppressed, or activated, and only on rare occasions, the same neurons showed both types of responses. Across a 30-fold concentration range, cortical neurons had diverse and non-monotonic responses, more neurons becoming suppressed with increases in concentration. The latency of the early responding principal cells (within 100 ms from odor onset) was generally not affected by changes in concentration, while late responders showed decreased average latency with increasing concentration. Expressing ChR2 in inhibitory interneurons (VGAT+), enabled the authors to determine that their response latency decreases with increasing concentration. Using three coding strategies (ensemble membership, summed spike count and binned spike count), the identity and concentration of odorants could be extracted with high accuracy by pooling together the ~450 recorded neurons. On average, for same number of cells included in the analysis, the decoding accuracy was higher for odor identity vs. intensity. Using a time-based coding strategy (binned spike count) substantially increased accuracy compared to a rate code (summed spike count) for intensity decoding, and only to a less degree for identity decoding. Using either an expanding or a sliding window strategy, the authors find that identity decoding accuracy is highest early after stimulus onset (~100ms), while intensity decoding accuracy is poor to begin with and steadily increases over the next 100 ms.

In my opinion this is a very interesting study that brings novel, and exciting information about cortical neurons responds to changes in odor identity and intensity. The experiments are carefully executed, and the data appears rich and of high quality.

However, I disagree with the current interpretation of the results and main message of the manuscript, as described by its title. I think the data presented here does not support some of the central claims made by the authors. I will explain below my concerns.

The claim for a temporal code to extract odor intensity is weak. The main result plotted in Figure 4 shows that the binned cumulative curve reaches higher accuracy with fewer cells compared to the summed count, or ensemble membership coding. However, even for identity decoding, the trend is the same (superior accuracy for binned spike count). The claim rests on the observation that including temporal information only marginally improves identity decoding but substantially improves intensity decoding. But to begin with, the classification accuracy for odor identity is superior for the same number of included cells, perhaps leading to a ceiling effect.

To support a strong claim regarding the existence of two complementary and multiplexed codes in which odor identity is represented by specific ensembles of odor responsive neurons (using a rate code) and odor intensity is encoded using spike time information, the evidence in the data has to indicate that a rate-code is better for decoding odor identity and a time-code is superior for concentration. Importantly, the authors find, in fact, that including timing information increases the performance for decoding both odor identity and concentration. So, why not conclude that time-based code is superior to rate-based one for decoding both concentration and identity?

To my mind, the data presented here indicates that the accuracy of concentration decoding (as assessed by this classifier) is poorer than odor identity decoding in the piriform cortex. However, accuracy of classification for both stimulus features can be improved by including time information.

The authors observe that the latency of early responding cells remains invariant to changes in concentration. In principle, this could be an important component of coding odor identity, irrespective of concentration. It suggests that temporal information (lack of change in response latency) is important for coding odor identity, since the early responding cells provide an invariant reference to a potential downstream decoder. Furthermore, this finding, together with changes in response latency of later responding cells imposes certain constraints on the validity of theories in the field on primacy coding for odor identity. To my mind, this is an important observation that is unfortunately not clearly emphasized in the current version of the manuscript.

*Reviewer #3:*

This manuscript by Bolding et al. provides an account of odor representations in piriform cortex using electrophysiological recordings. The manuscript demonstrates that both odor identity and concentration information is encoded in this network. Through the use of linear classification of neural activity vectors, the authors compare potential coding schemes that may be used by the brain to decode odor identity and make the case that odor identity and concentration information are best encoded using different schemes. Odor identity information is encoded well with a "membership" code that eschews fine temporal and spike count information, while temporal information is much more useful for decoding concentration. By classifying cropped activity vectors, they demonstrate that concentration information in their recorded population is available later following inhalation than information needed to decode odor identity. Finally, the authors demonstrate that inhibitory network elements are responsible for attenuating late responses, which they claim increases synchrony with increases in odor concentration.

Overall this work is an important addition to our understanding of olfactory cortex and it coding schemes. However, I have some issues that I feel must be addressed before it is published.

1) Classifier analysis is used to build an argument that late representations contain more concentration information. This argument is true only for the authors' recorded units and using linear classification. While this methodology is sufficient to demonstrate that information is present within the network, it is not sufficient to prove the absence, or relative absence, of information in the network. The authors should discuss this in the manuscript.

2) Piriform cortical representations are sparse. Thus at least two factors need to be considered in the analysis. First, the authors demonstrated that early activity is useful for generalizable odor identity by training the classifier on 3 concentrations and testing on the left-out concentration. Using this training/testing paradigm, a linear classifier will weight stable dimensions (i.e. neurons) over dimensions that are unreliable across concentrations. Only a small subset of neurons may be encoding odor identity despite large variance within the population. Can authors provide an analysis of the degree to which their classifier is using sparse weightings in this task? Can they identify the best performing neurons?

3) Second, the authors presented the dependence of the classifier performances as a function of number of neurons, n. The classification analysis may not accurately represent the information contained in the population. It is possible that a discriminator with input from specific combinations of n neurons performs much better than an average performance of a discriminator for the same number of neurons. What is the distribution of the discriminator performances for a given number of neurons? How the dependencies on the Figure 4 change for the best (or almost the best) n performing neurons, but not for average subset of n neurons.

4) While I agree that the fitting of a mixture of 3 Gaussians to the latency data looks reasonable by eye, I would ask that the authors provide more formal model comparison to justify the model and the number of mixed Gaussians. Once this is established, the authors should provide some estimate of the confidence in the fit parameters, as this confidence is important in establishing how the latencies shift between the olfactory bulb and piriform and whether these differences are significant. Bootstrapping across units can be used to derive these statistics.

5) The authors find that concentration information in their recorded population of units is encoded later than that of identity. At the end of the subsection “Dissociating representations of identity and intensity”, they claim that this conveniently matches behavioral observations from Abraham et al. Such a comparison of discriminant performance with a behavioral result seems like a stretch – the cited paper attributed the longer reaction time to the difficulty of the task, rather than to a difference between identity and concentration discriminations. In addition, the authors may consider citing Resulaj 2015, which reported very fast concentration discrimination.

6) More description of the methods used in analysis would be helpful. Latency determination for neural responses should be described, especially given that many of these neurons' firing rates are non-stationary within the sniff cycle. Also, PSTH convolution methodology is not described well – uniform kernel or Gaussian? Are non-overlapping (invalid) portions of the signal discarded, and if so are the timings adjusted for the shift that this would incur?

7) No methodology for olfactory bulb recordings is given in the manuscript. Some statistics should also be provided about the recorded population (i.e. recording depths, number of units recorded).

8) The authors provided an argument that inhibition is responsible for an increase of synchrony in the network (Figure 6). I did not understand this argument. My impression was, that the authors demonstrated that inhibition attenuated the later response, thus making the earlier response more temporarily confined. How does this lead to increase of synchrony in early response?

---

## [Author Response]

*Essential revisions:*

*1) The authors should analyze the concentration coding using a formal model that quantifies fit and confidence. This should address the observations of latency, inhibitory contributions, and synchrony.*

We have taken a formal model-fitting approach to determine an appropriate model for our latency distribution data. Our aim was to confirm and quantify the impression that two processes with differential concentration-sensitivity best explain the distribution of response latencies. However, when only two Gaussians are fit, extraneous, very late responses often distort the fit and pull the mean of the second Gaussian far to the right of the obvious peak in the histogram. A third stimulus-insensitive process can account for these late responses and improves the fits to the first and second peaks. This is now formally quantified by comparing the Bayesian Information Criterion (BIC), which quantifies goodness-of-fit while penalizing any increase in the number of free model parameters. Across multiple concentrations, bootstrap samples, and even across brain regions (i.e. piriform and bulb), a model with three Gaussians was identified as having the lowest BIC score. This justifies the use of three Gaussians in our mixture model.

Per reviewer 3’s suggestion, we used bootstrap analysis to establish confidence in fit parameters. We resampled our latency distributions 1000 times for each concentration and re-fit three Gaussian distributions to each bootstrap sample. These data show two populations of odor responses in piriform (Figure 4) and bulb (Figure 6) whose timing changes systematically with concentration; specifically, we find a population of fast responses whose timing only changes minimally with concentration and a slower population of responses whose latencies decrease substantially at higher concentration. Importantly, we point out that we can accurately fit the distribution of response latencies across concentration by changing only the mean and standard deviations of the first and second peak (i.e. the third Gaussian is invariant and we keep the proportion of responses in each phase constant).

We have now also measured and compared response latencies for populations of bulb and piriform neurons across concentrations within an experiment. Latency to first spike and latency to peak decreased in both bulb and piriform at higher concentrations (Figure 7). However, the steepness of this relationship is significantly greater in piriform than bulb. This result indicates that responses in piriform are not simply reflecting concentration-dependent changes in bulb.

Similarly, using the bootstrap data we determine 95% confidence intervals for the latencies of the first and second response. We find that these can be accurately predicted by spiking of piriform inhibitory interneurons. This finding considerably bolsters our conclusion that intracortical inhibition plays an important role in shaping piriform odor responses.

We have also elaborated and clarified our argument, and provided additional data and analyses, to support our finding that synchrony increases with concentration, as described in answer to reviewer 1 and 3’s comments below.

*2) The classifier analysis should be should be extended to establish whether the coding is sparse.*

This question has forced us to go back and look at our data, question our starting assumptions, and forced us to arrive at a different conclusion. Per reviewer 3’s suggestion, we examined the distribution of classifier accuracies achieved using different, random subsets of cells. If coding is sparse then most cells should provide little or no information about the stimulus. In this case, the distribution of classifier accuracies should be heavily skewed when classifying with small populations; classification should mostly be very poor because most cells do not respond to the few odors presented and occasionally be very good if just the right cells are selected. Instead, we see a normal distribution of accuracies (Figure 5), which indicates that most cells provide some information about the stimulus. We also find that we can almost always achieve relatively good classification (>80% accuracy) with only ~70 cells (>97.5% confidence).

We note that this contradicts our initial claim. We previously said: “In summary, we found that different odors activated distinct, sparse and overlapping ensembles of piriform cortex neurons.” We had used this term somewhat loosely, based on our measures of lifetime and population sparseness. In fact, our sparseness measures were slightly lower than Miura et al.’s (2012), who described representations as “moderately sparse”. In fact, we think that determining whether or not a representation is sparse by the ability to decode using small and random subsets of neurons is more appropriate than measures of lifetime and population sparseness. According to this analysis, we conclude that odor representations in piriform cortex are *not*sparse. We specifically thank reviewer 3 for forcing us to revisit and rethink this important point. We have changed the text in our Results and added to the Discussion to address this point.

*3) The authors should revisit the temporal analysis and conclusions, and clearly state how strong is the confidence in them.*

We have performed additional analyses of the temporal properties of responses and the role of spike-time information in cortical odor coding. We can confidently make the following statements (i.e. findings are statistically significant):

1) Response latencies decrease systematically at higher concentrations (Figure 4).

2) The distribution of piriform responses can be described by two phases: a rapid and largely concentration-invariant phase and a slower phase whose latencies decrease with concentration (Figure 4, Figure 4—figure supplement 1).

3) Responses are more synchronous at higher concentrations (Figure 3, Figure 4).

4) Spike time information does not improve decoding accuracy for odorant classification (Figure 5, Figure 5—figure supplement 1) or odor identity generalization (Figure 5; Figure 5—figure supplement 1).

5) Spike time information significantly improves odor concentration classification (Figure 5; Figure 5—figure supplement 1).

6) The earliest responses provide little information about odor concentration (Figure 6).

7) The distribution of responses in bulb can also be described by a rapid, largely concentration- invariant phase and a slower concentration-dependent phase (Figure 7, Figure 7—figure supplement 1).

8) Response latencies are more steeply concentration-dependent in piriform than bulb (Figure 8).

9) The distribution of response latencies can be predicted by the spiking of piriform interneurons (Figure 9).

*4) Some details of methodology need further elaboration: e.g., analysis, bulb recordings, and unit isolation.*

We have included additional information in the Methods section that describes our analyses procedures as requested by the reviewers, details about bulb recordings, and provided spike sorting statistics including the distribution of isolation distances and refractory period violations for all units recorded in bulb and cortex.

*The outcome of the analysis above may suggest that the claims in the title should be reconsidered.*

We have carefully considered the reviewers’ concerns and, now with additional analyses, stand by our original claim that there are complementary codes for representing odor identity and intensity.

Reviewer 2 makes the following argument:

“To support a strong claim regarding the existence of two complementary and multiplexed codes […] the evidence in the data has to indicate that a rate-code is better for decoding odor identity and a time-code is superior for concentration. Importantly, the authors find, in fact, that including timing information increases the performance for decoding both odor identity and concentration. So, why not conclude that time-based code is superior to rate-based one for decoding both concentration and identity?”

There are two issues here: whether spike time information is used for identity coding and whether a time code is better than a rate code for intensity coding. First, reviewer 2 is pointing to what appears to be slightly better average decoding accuracy for odorant classification using binned vs. summed response vectors (Figure 5). However, as we state in the text, the generalization task, in which the decoder must “identify a familiar odor at a novel concentration”, provides a better way of testing an odor identity code than simply classifying responses to different odorants at a nominal concentration, and here the average performances overlap (Figure 5). Nevertheless, we have now added additional analyses that show that differences in classification accuracy using different coding schemes never reach statistical significance (for this decoder) for either odorant classification or generalization. In fact, we see a trend for worse generalization for odor identity when spike time information is used. Therefore, we stand by our original point that an ensemble code is sufficient for encoding odor identity with little if any additional information provided by spike count or spike time information.

Second, we also find almost no statistically significant improvement in decoding accuracy when we add spike rate information to a membership code (Figure 5, Figure 5—figure supplement 1). However, adding spike time information to a rate code significantly increases decoding accuracy for pseudopopulations >250 cells (Figure 5, Figure 5—figure supplement 1). We feel this statistical analysis therefore indicates that a time-based code *is* superior to a rate-based code for representing concentration. We therefore feel confident claiming that there are complementary codes for odor identity and odor intensity in piriform cortex.

However, we do note that a membership code does provide considerable information about odor concentration. We therefore do not claim that these codes are independent or completely multiplexed, and we address this point in the Discussion.

*Additional specific comments are also indicated in individual reviewer comments below.*

*Reviewer #1:*

*[…] The authors have a key result on concentration-dependent coding. This is valuable but could be sharpened. There is quite a bit of data here which is potentially very informative, however the analysis could probably be taken a bit further.*

We have extended our analysis according to, and beyond, the suggestion of all three reviewers. We think this strengthens the manuscript considerably, and we thank the reviewers for their helpful suggestions.

*Then the authors pursue an analysis of concentration-dependent latency. There are two aspects to this, both of which could be strengthened.*

*1) First, can the concentration-dependence be explained simply by bulbar concentration-dependent latency? To their credit, the authors have directly recorded and shown the bulbar concentration-dependence. At present it isn't clear if the piriform is doing any substantial transformation of the response.*

We have extended our data analysis to clearly show that bulb input is actively transformed in piriform cortex. First, we have added a figure that overlays the distributions of response latencies in bulb and piriform at different concentrations (Figure 8). If piriform cells were simply following the bulb, then these distributions should overlap substantially. We see a rapid peak in bulb responses that is reflected in piriform. However, bulb cells continue to respond throughout the sniff, consistent with previous findings (Bathellier et al., 2008; Cury & Uchida, 2010; Shusterman et al., 2011). By contrast, there is a marked decrease in later piriform responses (100-300 ms after inhalation) in piriform versus bulb. This result indicates that early bulb inputs are better able to drive responses in piriform cortex than later inputs, and the most parsimonious explanation for this suppression is inhibition within piriform cortex. Therefore, our data show that piriform cortex is substantially transforming input from bulb.

Next, we specifically demonstrate that the concentration-dependent change in response timing is significantly steeper in cortex than bulb. This is true using either the distribution of latencies to first spike (Figure 8) or latencies to peak (Figure 8) for each cell. Figure 8 shows data from an example experiment with simultaneously recorded populations of bulb and piriform cells. (All 6 populations of bulb recordings have paired piriform recordings but 3 of the 9 piriform populations were recorded alone. Data are pooled across experiments in Figure 8.) Again, if piriform cortex were simply following input from bulb then these should change together. In fact, our data suggest that the second phase of piriform responses peaks slightly *before* the second phase in bulb, although this did not reach statistical significance, even with the bootstrapping analysis.

*2) There is a nice experiment wherein the authors investigate the mechanisms behind concentration coding by optogenetically targeting inhibitory interneurons and delivering light pulses to identify the neurons from the recorded dataset. They characterized the responses of these neurons. Using this, the authors seem to have an implicit model whereby the inhibitory timing in the piriform acts to separate out the second peak in excitatory neurons from the first one. This implies that concentration dependence of the inhibitory peaks (Figure 6) should account for the shift in second peak timing for the remaining cells (Figure 6, Figure 3). Does such a model work? Does it improve on the already present concentration-dependent latency response from the bulb? Whether or not my interpretation is what the authors had in mind, it would be useful for the authors to make their response model explicit, and test it.*

The reviewer’s interpretation is mostly what we have in mind. That is, we propose that the concentration-dependence of the inhibitory response (i.e. shorter latencies and more synchronous spiking at higher concentrations, Figure 9) plays a role in shifting the timing of the second phase of the response in the remaining cells. As per the reviewer’s suggestion, we now show inhibitory spiking as curves instead of heatmaps (Figure 9) and we have overlaid these with the 2.5-97.5% confidence intervals for times of the first and second peak derived from the bootstrapping analysis suggested by reviewer 3. This representation clearly indicates how inhibition peaks shortly after the peak of the first phase and that the second phase of the response peaks during the subsequent trough in inhibition. These data suggest that shifts in the timing of inhibition partially account for the shift in timing of the cortical responses.

However, we do not claim that the timing of the inhibition can fully account for the shift in response latencies. We tried convolving bulb output with the spiking of VGAT+ cells but could not accurately predict either the distribution of piriform response times or the population PSTH (not shown). This is not surprising. Such a convolution should work in a simple feedforward circuit in which both piriform cells and inhibitory interneurons were only, or largely, driven by bulb input. However, principal cells in piriform cortex are interconnected by extensive recurrent collateral connections that also drive strong feedback inhibition, and we (Franks et al., 2011) and others (Davison & Ehlers, 2011; Poo & Isaacson, 2011; Large et al., 2016) have shown that recurrent/intracortical excitation and feedback inhibition can strongly and dynamically shape piriform output. Consequently, a considerably more complex model would be required to accurately predict piriform spiking from bulb output. We now state this point explicitly in the manuscript. A detailed spiking network model that captures the statistics of bulb output and the specific organization of piriform circuitry can shed more light on how this transformation is implemented (Stern, Abbott, Franks; Cosyne, 2014; Stern, Bolding, Abbott, Franks, Cosyne, 2015; Stern et al., in prep.), but this is beyond the scope of the present study.

*Reviewer #2:*

*[…] However, I disagree with the current interpretation of the results and main message of the manuscript, as described by its title. I think the data presented here does not support some of the central claims made by the authors. I will explain below my concerns.*

*The claim for a temporal code to extract odor intensity is weak. The main result plotted in Figure 4 shows that the binned cumulative curve reaches higher accuracy with fewer cells compared to the summed count, or ensemble membership coding. However, even for identity decoding, the trend is the same (superior accuracy for binned spike count). The claim rests on the observation that including temporal information only marginally improves identity decoding but substantially improves intensity decoding. But to begin with, the classification accuracy for odor identity is superior for the same number of included cells, perhaps leading to a ceiling effect.*

*To support a strong claim regarding the existence of two complementary and multiplexed codes in which odor identity is represented by specific ensembles of odor responsive neurons (using a rate code) and odor intensity is encoded using spike time information, the evidence in the data has to indicate that a rate-code is better for decoding odor identity and a time-code is superior for concentration. Importantly, the authors find, in fact, that including timing information increases the performance for decoding both odor identity and concentration. So, why not conclude that time-based code is superior to rate-based one for decoding both concentration and identity?*

*To my mind, the data presented here indicates that the accuracy of concentration decoding (as assessed by this classifier) is poorer than odor identity decoding in the piriform cortex. However, accuracy of classification for both stimulus features can be improved by including time information.*

We have addressed this question above. In summary, odor identity coding is best tested using the generalization task, in which decoding accuracy is equivalent using binary, summed and binned response vectors. Moreover, we found no statistically significant differences between decoding accuracy using these different types of input for either generalization or simple odorant classification. We therefore conclude that our data indicate spike time information does not contribute to odor identity coding. Second, we show that there is almost no significant improvement in decoding accuracy when spike count information is added to a membership code, but considerable improvement when spike time information is added to a rate code, which indicates that spike time information is superior to spike count for intensity coding.

*The authors observe that the latency of early responding cells remains invariant to changes in concentration. In principle, this could be an important component of coding odor identity, irrespective of concentration. It suggests that temporal information (lack of change in response latency) is important for coding odor identity, since the early responding cells provide an invariant reference to a potential downstream decoder. Furthermore, this finding, together with changes in response latency of later responding cells imposes certain constraints on the validity of theories in the field on primacy coding for odor identity. To my mind, this is an important observation that is unfortunately not clearly emphasized in the current version of the manuscript.*

We are glad the reviewer agrees that these observations are important and we have emphasized them more clearly in both the Results and Discussion sections. We have also attempted to discuss them more clearly in the context of a primacy code.

*Reviewer #3:*

*[…] Overall this work is an important addition to our understanding of olfactory cortex and it coding schemes. However, I have some issues that I feel must be addressed before it is published.*

*1) Classifier analysis is used to build an argument that late representations contain more concentration information. This argument is true only for the authors' recorded units and using linear classification. While this methodology is sufficient to demonstrate that information is present within the network, it is not sufficient to prove the absence, or relative absence, of information in the network. The authors should discuss this in the manuscript.*

While we suspect that the ~500 cells we recorded from are reasonably representative, we concede that this constitutes less than 1% of the piriform cells in any one animal and that there may well be cells that are early encoders of concentration, or many other odor features. We now discuss this in the manuscript, stating that other cells, other coding strategies, or even other brain areas may decode olfactory information differently, allowing, for example, early intensity representations.

*2) Piriform cortical representations are sparse. Thus at least two factors need to be considered in the analysis. First, the authors demonstrated that early activity is useful for generalizable odor identity by training the classifier on 3 concentrations and testing on the left-out concentration. Using this training/testing paradigm, a linear classifier will weight stable dimensions (i.e. neurons) over dimensions that are unreliable across concentrations. Only a small subset of neurons may be encoding odor identity despite large variance within the population. Can authors provide an analysis of the degree to which their classifier is using sparse weightings in this task? Can they identify the best performing neurons?*

We have discussed the issue of sparseness in the general comments above. With respect to this specific comment, in the generalization task the classifier was trained on stimuli individually and the three concentrations of the target odor were not lumped together until after classification. So, the classifier does not know which dimensions are stable across stimuli.

*3) Second, the authors presented the dependence of the classifier performances as a function of number of neurons, n. The classification analysis may not accurately represent the information contained in the population. It is possible that a discriminator with input from specific combinations of n neurons performs much better than an average performance of a discriminator for the same number of neurons. What is the distribution of the discriminator performances for a given number of neurons? How the dependencies on the Figure 4 change for the best (or almost the best) n performing neurons, but not for average subset of n neurons.*

This is an excellent suggestion and we have followed it to provide an important clarification on sparseness of cortical odor representations. Please see above.

*4) While I agree that the fitting of a mixture of 3 Gaussians to the latency data looks reasonable by eye, I would ask that the authors provide more formal model comparison to justify the model and the number of mixed Gaussians. Once this is established, the authors should provide some estimate of the confidence in the fit parameters, as this confidence is important in establishing how the latencies shift between the olfactory bulb and piriform and whether these differences are significant. Bootstrapping across units can be used to derive these statistics.*

We have formalized our justification for 3 Gaussians using the Bayesian Information Criterion, and then provided confidence for these conclusions. Please see above.

*5) The authors find that concentration information in their recorded population of units is encoded later than that of identity. At the end of the subsection “Dissociating representations of identity and intensity”, they claim that this conveniently matches behavioral observations from Abraham et al. Such a comparison of discriminant performance with a behavioral result seems like a stretch – the cited paper attributed the longer reaction time to the difficulty of the task, rather than to a difference between identity and concentration discriminations. In addition, the authors may consider citing Resulaj 2015, which reported very fast concentration discrimination.*

We find that early piriform responses are relatively concentration-invariant over the first 100 ms before they become more concentration-dependent. Note that this is not only true for the discriminant analysis but also for the actual spiking data (Figure 3). We therefore think it is valid to propose that the slower evolution of concentration-dependent differences in piriform responses can account for at least some of the difference in reaction times. We nevertheless agree with the reviewer that it is a stretch to attribute differences in reaction times entirely to differences in piriform odor responses and we have softened our statement to say that this difference in cortical representations may account for some of the differences in behavior, and we have cited the Resulaj paper as a counter-example.

*6) More description of the methods used in analysis would be helpful. Latency determination for neural responses should be described, especially given that many of these neurons' firing rates are non-stationary within the sniff cycle. Also, PSTH convolution methodology is not described well – uniform kernel or Gaussian? Are non-overlapping (invalid) portions of the signal discarded, and if so are the timings adjusted for the shift that this would incur?*

The following paragraph from the Methods section states that we used a Gaussian kernel for smoothing:

“We computed smoothed kernel density functions (KDF) with a 10 ms Gaussian kernel (using the psth routine from the Chronux toolbox: www.chronux.org) to visualize trial-averaged firing rates as a function of time from inhalation onset and to define response latencies for each cell-odor pair. Peak latency was defined as the maximum of the KDF within a 500-ms response window following inhalation.”

The resulting KDF reflects only spike times that occurred within the boundaries of the response window, but is computed over a wide enough time span to accommodate the spread of Gaussian kernels near the edges of the window. We have adjusted our description to clarify this:

“To define peak latencies, KDFs were computed from spike times occurring with a 500-ms response window following inhalation. Peak latency was the time of maximum of this KDF. In cases where the KDF maximum was at the window edge, indicating falling from or rising to a peak outside of the response window, peak latency was undefined.”

*7) No methodology for olfactory bulb recordings is given in the manuscript. Some statistics should also be provided about the recorded population (i.e. recording depths, number of units recorded).*

We have provided further descriptions, including the recording depths and total number of isolated units.

*8) The authors provided an argument that inhibition is responsible for an increase of synchrony in the network (Figure 6). I did not understand this argument. My impression was, that the authors demonstrated that inhibition attenuated the later response, thus making the earlier response more temporarily confined. How does this lead to increase of synchrony in early response?*

We apologize for lack of clarity here. As we discuss above, we are quantifying synchrony here as the distribution of response times over the 500 ms sniff. Therefore, and consistent with the reviewer’s impression, as increasing inhibition at higher concentrations attenuates more of the later response a greater fraction of the total response occurs earlier. We are not defining population synchrony here as synchrony of the early response specifically. Nevertheless, it is important to note that there is indeed an increase in the synchrony of the early response, and that this could be responsible for stronger and more synchronous recruitment of inhibition. In principle, this increased early synchrony could be used to encode concentration, however our sliding-window decoding analysis does not support this. Importantly, and to the reviewer’s question, we are not claiming a role for inhibition in synchrony of the early response.